# Preparation and Properties of a Sulphoaluminate Magnesium-Potassium Phosphate Green Cementitious Composite Material from Industrial Solid Wastes

**DOI:** 10.3390/ma14237340

**Published:** 2021-11-30

**Authors:** Changzai Ren, Wenlong Wang, Dongliang Hua, Shuang Wu, Yonggang Yao

**Affiliations:** 1School of Energy and Power Engineering, Qilu University of Technology, Jinan 250306, China; huadl@qlu.edu.cn; 2National Engineering Laboratory for Coal-Fired Pollutants Emission Reduction, Shandong University, Jinan 250061, China; wwenlong@sdu.edu.cn (W.W.); wushuanglc@163.com (S.W.); youxiatianxia2008@163.com (Y.Y.)

**Keywords:** industrial solid waste, sulphoaluminate–magnesium–potassium–phosphate cementitious composite material, clinker phase, water resistance, sustainable development

## Abstract

The preparation of high-performance green cementitious material from industrial solid waste is a feasible large-scale utilization approach for industrial solid waste. This work investigates the feasibility of using industrial solid wastes in a sulphoaluminate–magnesium–potassium–phosphate cementitious composite material (SAC-MKPC) clinker preparation and the influence of the calcination temperature and clinker ingredients on the hydration behavior and mechanisms of the SAC-MKPC with a Mg/P ratio of 5. The results show that the novel SAC-MKPC that was prepared from aluminum slag, carbide slag, coal gangue, and magnesium desulfurization slag was composed mainly of mineral MgO, $C_{4}A_{3}\bar{S}$, and C_2_S and the calcination temperature of the main mineral phases was 1250–1350 °C. The solid-waste-based SAC-MKPC had better mechanical properties and excellent water resistance compared with the MKPC. The optimal compressive strength reached 35.2, 70.9, 84.1, 87.7, and 101.6 MPa at 2 h, 1 d, 3 d, 7 d, and 28 d of hydration, respectively. The X-ray diffraction spectra and scanning electron micrographs of the hydration products of the SAC-MKPC clinker showed that AFt and K-struvite crystals coexisted and adhered to form a dense structure. This work provides an innovative idea to produce green cementitious material using industrial solid wastes and may promote the sustainable development of the power and mining industries.

## 1. Introduction

The electrolytic aluminum, power, and mining industries produce large volumes of solid waste [1]. The industrial solid waste is dumped in huge volumes over a wide geographical distribution, which causes far-reaching environmental harm in China [2]. The preparation of eco-friendly construction materials from industrial solid waste is a feasible large-scale utilization method using current technologies [3]. Magnesium-potassium phosphate cement (MKPC) is an excellent air-hardening cementitious material with a wide range of applications, including urgent construction and repair, nuclear waste encapsulation, and three-dimensional printing building materials; however, the high energy consumption and poor water resistance of MKPC have constrained its application in engineering [4,5,6]. To improve MKPC’s water stability and low carbon emission capability, extensive theoretical and experimental research has been conducted in three main areas, namely filling theory, interaction theory, and coating theory. In filling experiments and theoretical research, fly ash, nickel slag, and silica fume have been used as filling materials owing to their beneficial effects on improving the fresh mixture’s workability and reducing the development of heat in MKPC-based blends, which led to a more compact microstructure of hardened MKPC paste [7,8,9]. In interaction experiments and theories, granulated blast furnace slag and metakaolin have been used as reactive materials in MKPC-based blends [10]. The two materials blend with MKPC to form a new amorphous phase, comprised of aluminum phosphate and calcium phosphate, which prevents water erosion [11]. In coating experiments and theories, amorphous calcium sulfoaluminate has been used as a coating material. Hydration products that were coated on the MKPC surface prevent water from entering the MKPC paste and improve the water resistance. A novel approach to improve the water resistance, reduce the use of natural resources, and maintain or improve the performance of MKPC is required.

The novel sulphoaluminate magnesium–potassium phosphate cementitious composite material (SAC-MKPC) was first proposed in 2017, and it exhibits excellent mechanical stability and favorable water stability [12,13]. The current findings suggest that unreacted SAC clinker particles and the hydration product have three functions (a filling effect, an interaction effect, and a coating effect) and improve the water resistance and performance. The novel SAC-MKPC exhibited better mechanical properties and excellent water resistance, in which the MKPC was composed of dead-burnt magnesia and potassium dihydrogen phosphate (KH_2_PO_4_, KDP). However, dead-burnt magnesia needs to be calcined at 1700 °C for 7–10 h, which requires more energy [14], and the high energy consumption was found to be the greatest disadvantage of the SAC-MKPC. We investigated the possibility of producing SAC-MKPC clinker from magnesium sulphate, calcium oxide, and aluminium oxide in different proportions, at low calcination temperatures, and using chemical reagents as raw materials [15].

To decrease the environmental pollution caused by industrial solid waste and to improve the added value of solid-waste-based products, a method of preparing SAC-MKPC clinker with typical industrial solid waste was proposed. Based on an analysis of a SAC and MKPC cement system, this study explores the possibility of producing SAC-MKPC clinker from a mixture of aluminum slag, carbide slag, coal gangue, and magnesium desulfurization slag at a low calcination temperature. The objective of this study is to determine the SAC-MKPC clinker’s formation mechanism, hydration behavior, mechanical properties, and hydration products, the water resistance of the hydration products, and the microstructure of SAC-MKPC composites by X-ray diffraction (XRD), thermal gravity–differential thermal gravity (TG-DTG) analysis, and scanning electron microscopy–energy dispersive spectrometry (SEM-EDS).

## 2. Materials and Methods

### 2.1. Experimental Instruments and Testing Methods

#### 2.1.1. Experimental Instruments

The experiment was divided into three categories: raw material preparation, SAC-MAPC clinker preparation, and performance testing. The main pieces of experimental equipment were divided into raw material preparation equipment, clinker calcination and molding equipment, and analytical equipment. The main experimental equipment were shown as Table 1.

During the raw material’s preparation, an electronic balance, a disc refiner, a sample pulverizer, a hot air oven, and a box resistance furnace were used to dry, weigh, grind, and homogenize the raw materials. During the clinker calcination and molding process, a standard sieve, a steel mold, and a cement paste vibration table were used to grind and mold the standard test blocks. In terms of the analytical performance of the clinker and hydration products, an automatic pressure measurement testing machine, XRF, and XRD were used to study the mechanical properties, mineralogical compositions, and microstructure of the hydrated products, respectively. Thermal gravimetric analysis was used to determine the category and degree of the SAC-MKPC specimen’s hydration products. 

#### 2.1.2. Testing Methods

A.Initial setting time

At an ambient temperature of 25 °C, the setting time of the fresh SAC-MKPC paste was measured with the automatic setting time tester according to GB/T1346-2011 [16]. Since the time between the initial and the final setting was very short (several minutes), only the initial setting time is presented in this paper.

B.Specimen preparation and compressive strength

The SAC-MKPC paste was cast into 20-mm cubic molds, and molds with SAC-MKPC paste were cured in a curing box at a constant temperature and humidity of 25 °C and 55%, respectively. After a curing time of 1 h, the standard test block was demolded and cured in a standard curing box under the same conditions.

The compressive strength of the samples was measured using an automatic pressure measurement testing machine. Each proportion was measured as a set of two samples. The samples, after air curing for 2 h, 1 day, 3 days, 7 days, and 28 days, were tested for compressive strength. To test the water resistance, the compressive strength of the samples was measured after 28 days in water. Before testing, the samples immersed in water were taken out of the water and dried for 4 h.

C.Thermal gravity–differential thermal gravity (TG-DTG) analysis

In the thermal mass tests, the mass of the test paste powder was ~20 mg, and the temperature range of the thermogravimetry test was 30–700 °C at 10 °C/min. TG-DTG analyses were used to investigate the mineral composition phases of the SAC-MKPC hydrated product.

D.XRD analysis

The mineral composition phases of the SAC-MKPC clinker and hydration products were identified by XRD with a Ni filter and Cu–Ka (k = 1.5406 Ǻ) radiation at 40 kV and 30 mA and at a scanning speed of 0.02°/s over a 2 h range and 10–90° with 4000 data points. A quantitative analysis of the main phase was conducted by using the “TOPAS” software package and the fitted parameters included the background coefficients, instrument parameters, sample corrections, cell parameters, and phase-shape parameters. The pseudo-Voigt function, structure, and Inorganic Crystal Structure Database (ICSD) codes used for the Rietveld refinement were based on our previous research [17].

E.Microstructure and morphology

Hydration samples were collected from the middle sections of the SAC-MKPC paste specimens and immersed in anhydrous alcohol for at least two days to stop further hydration. The samples were vacuum-dried at 60 °C for 24 h. The morphologies of the hydration products in the hardened MKPC paste were analyzed by using a QUANTA-200 environmental scanning electron microscope and chemical characterization was performed using EDS. The samples were coated with gold before examination to improve the conductivity.

### 2.2. Materials

Industrial solid wastes (aluminum slag, carbide slag, coal gangue, and magnesium desulfurization slag (MDS)) were used to prepare SAC-MKPC clinkers. Magnesium desulfurization slag was collected from Weiqiao coal-fired power plants. Aluminum slag was supplied by Weiqiao Pioneering, Shandong. Carbide slag was obtained from Liaocheng acetylene company, and coal gangue was obtained from Taifeng Group Ming in Taian, Shandong. The main oxide compositions of the initial raw materials were analyzed chemically by wavelength-dispersive X-ray fluorescence (XRF). Because of the release of gaseous ammonia from the MgO-NH_4_H_2_PO_4_ system, KH_2_PO_4_ (KDP) was one of the main raw materials in the SAC-MKPC preparation. The chemical composition of the raw materials is presented in Table 2. 

The magnesium desulfurization slag was formed by SO_2_ absorption by light-burned magnesium oxide in a desulfurization tower. The XRD pattern of the magnesium desulfurization slag is shown in Figure 1a. The magnesium desulfurization slag was mainly composed of magnesium sulfite hexahydrate (MgSO_3_·6H_2_O), magnesium sulfate heptahydrate (MgSO_4_·7H_2_O), calcium oxide (CaO), and magnesium hydroxide (Mg (OH)_2_). MgSO_3_·6H_2_O, MgSO_4_·7H_2_O, and Mg(OH)_2_ are the most important major phases in magnesium desulfurization slag, and these phases comprise 95% of the magnesium desulfurization slag material. Carbide slag is an industrial waste that is generated from the hydrolysis of calcium carbide (CaC_2_) during acetylene production [18]. Figure 1b presents the XRD pattern of carbide slag, where the main phases are calcium hydroxide (Ca(OH)_2_), calcium carbonate (CaCO_3_), and magnesium oxide (MgO). Silicon dioxide (SiO_2_) and ferric oxide (Fe_2_O_3_) are the most important minor phases in carbide slag. Ca(OH)_2_, CaCO_3_, and MgO are the most important major phases in the carbide slag and comprise 92% of the carbide slag phase. Aluminum slag is generated by the secondary aluminum industry. Figure 1c presents the XRD patterns of aluminum slag. It contains aluminum and aluminum oxide, and may include many impurities, such as sodium chloride, potassium chloride, calcium fluoride, sodium fluoride, aluminum, aluminum nitride, magnesium chlorides, and some heavy metals [1]. The major mineral phase contents of Al, Al_2_O_3_, AlN, and SiO_2_ comprise ~90% of the main components. Coal gangue is the tailing that is produced by coal mining. Figure 1d presents the XRD pattern of coal gangue. The main phases are silicon dioxide (SiO_2_), aluminum oxide (Al_2_O_3_), and magnesium oxide (MgO). Some minor-phase CaO, TiO_2_, and potassium salts are present and the main phases account for more than 80% of the content.

Figure 2a shows the TG-DTG outputs of the magnesium desulfurization slag. The MDS decomposition occurred in three main stages. MgSO_3_·6H_2_O and MgSO_4_·7H_2_O began to lose crystalline water from 150 to 230 °C. When the temperature exceeded 230 °C, MgSO_3_·6H_2_O and MgSO_4_·7H_2_O lost all of their crystalline water and transformed to anhydrous MgSO_3_ and MgSO_4_. When the temperature reached 430 °C, the MgSO_3_ began to decompose and form MgO and SO_2_, and the decomposition was completed at 550 °C, with a heat loss of 31%. When the temperature exceeded 900 °C, the MgSO_4_ began to decompose and transform into MgO, O_2_, and SO_2_. When the temperature exceeded 1060 °C, the magnesium sulfate was decomposed completely, the thermal mass loss reached 65%, the quality remained unchanged, and the thermal decomposition was complete. Figure 2b shows the TG-DTG outputs of the carbide slag. Two tiny peaks appear on the DTG curve at 80 °C and 115 °C and are caused mainly by the loss of free and crystalline water of the carbide slag [19]. When the temperature reached 400 °C, the Ca(OH)_2_ began to decompose and converted into H_2_O and CaO, and the decomposition was completed at 500 °C. As the temperature continued to increase, the CaCO_3_ began to decompose and generated CaO and released CO_2_ between 600 and 750 °C. Hence, the main quality loss for the carbide slag was the decomposition of Ca(OH)_2_ and CaCO_3_.

### 2.3. Experimental Methods

The SAC-MKPC cementitious composite material was prepared in the laboratory. The main experimental steps included raw mix preparation, calcining, secondary mixing, and molding.

(1)Raw mix preparation

Based on an analysis of the XRD patterns and TG-DTG outputs of the main raw materials and previous research (Section 2.1), the main possible chemical reactions of the solid-state materials were as follows.

(a)Calcium sulfate, magnesium oxide, and calcium oxide formed through the solid-state reactions of magnesium desulfurized slag and carbide slag [20].
(1)$${\mathrm{Ca}(\mathrm{OH})}_{2}+{\mathrm{CaCO}}_{3}+{\mathrm{MgSO}}_{3}+{\mathrm{MgSO}}_{4}\to {\mathrm{CaSO}}_{4}+\mathrm{MgO}+\mathrm{CaO}+{\mathrm{SO}}_{2}↑+O_{2}↑+{\mathrm{CO}}_{2}↑+H_{2}O$$
(b)Calcium oxide, aluminium oxide, and calcium sulfate transformed into the ye’elimite phase.
3CaO + 3Al_2_O_3_ + CaSO_4_ → 3CaO·3Al_2_O_3_·CaSO_4_(2)(c)Magnesium oxide transformed into dead-burnt magnesium oxide [21].
MgO→dead-burnt MgO(3)

Based on the above chemical reaction mechanism, the main possible reaction products were dead-burnt MgO, CaSO_4,_ and 3CaO·3Al_2_O_3_·CaSO_4_ ($C_{4}A_{3}\bar{S}$) in the SAC-MKPC clinker system. Six different raw material ratio schemes were designed. MgO had a theoretical 40%, 50%, 60%, 70%, 80%, and 100% share of the SAC-MKPC clinker system, respectively. Table 3 shows the proportions of raw materials and the calcination conditions. 

(2)According to the raw mixes in Table 2, a raw material block was prepared by mixing the raw materials and breaking, grinding, pressing, and drying the block. The dried raw material samples were calcined under electrical resistance, and the calcination temperature curve is shown in Figure 3. Mineral phases of the raw materials lost crystalline water below 400 °C, and, hence, the temperature increment was 8 °C/min. The decomposition reaction of calcium hydroxide and magnesium sulfite hexahydrate occurred from 400 to 800 °C and the temperature increment was set at 5 °C/min to achieve a complete reaction. The decomposition reaction of calcium carbonate and magnesium sulfate occurred, and the transient phase of calcium sulfate and gehlenite formed, at 800–1100 °C; hence, the rate of temperature increase was 4 °C/min. When the temperature exceeded 1100 °C, the calcined temperature increased from room temperature to the set point temperature and 3CaO·3Al_2_O_3_·CaSO_4_ and dead-burnt MgO formed. The temperature increment was 5 °C/min and the final temperature was maintained for 30 min. After calcination, the samples were taken out of the furnace and allowed to cool naturally to room temperature in air.

(3)Secondary mixing method

The hydration products of the SAC-MKPC cementitious composite material were comprised mostly of a magnesium-oxide particle skeleton, potassium phosphate, and ettringite. The main mineral phases included dead-burnt MgO, $C_{4}A_{3}\bar{S}$, and calcium sulphate for the SAC-MKPC clinker system. Hence, the SAC-MKPC cementitious composite material required secondary mixing with SAC-MKPC clinker, gypsum, and salt phosphate. According to previous research [22,23,24,25,26], the performance of the SAC-MKPC cementitious composite material depends on the magnesia-to-phosphate molar ratio (M/P), the phosphate type, and the water-to-binder ratio. For the two single systems, MKPC was prepared with a water/binder ratio (w/b) of 0.16, and SAC was prepared with a w/b of 0.28 [12,27]. Each addition of a SAC-MKPC clinker sample was to the same M/P. The M/P was set to 5/1, and gypsum of 5% mass SAC was added [28]. Therefore, for the SAC-MKPC system, the w/b mass ratio was 16% of the binder (where the binder was the total of MgO and KH_2_PO_4_) and 28% of the SAC. Several different water requirements were used, including 29.4, 29.6, 29.7, 29.8, and 29.9 (g/100 g) when the theoretical MgO contents were 40 wt.%, 50 wt.%, 60 wt.%, 70 wt.%, and 80 wt.%, respectively. The SAC-MKPC clinker was prepared according to the experimental schemes shown in Table 2 and the calcination temperature increase control curve shown in Figure 3. Table 3 shows the chemical components of the different SAC-MKPC clinker samples as measured by XRF, and the additive amount of KH_2_PO_4_, gypsum, and water for all clinker samples was estimated according to the XRF data. The amounts of added KH_2_PO_4_, water, and gypsum are listed in Table 4.

(4)Grinding and molding

SAC-MKPC clinker samples were pulverized by a clinker grinding mill, and the clinker powder’s diameter was controlled to less than 45 um. The SAC-MKPC cementitious composite material was mixed with water to form a slurry, and the slurry was poured into a standard mold (20 mm × 20 mm × 20 mm). Standard test blocks were prepared and cured according to Chinese national standard GB50204-2002 [29]. A batch of six 20 mm × 20 mm × 20 mm paste specimens were made for compressive strength testing.

## 3. Results and Discussion

### 3.1. Mineralogical Composition of the SAC-MKPC Clinker

The theoretical MgO content and calcination temperature are two key factors in the clinker and hydration properties of the paste, and they affect the physical and mechanical properties. The XRD patterns of the SAC-MKPC clinker system at different calcination temperatures and raw material ratios are shown in Figure 4, Figure 5, Figure 6 and Figure 7. Figure 4, Figure 5, Figure 6 and Figure 7 present the XRD patterns of the different theoretical MgO contents in the SAC-MKPC clinker system at the same calcination temperatures. The final clinker was mainly composed of ye’elimite ($C_{4}A_{3}\bar{S}$), periclase (MgO), and dicalcium silicate (C_2_S). A small amount of magnesia aluminum spinel (MgAl_2_O_4_), anhydrite (CaSO_4_), and akermanite (Ca_3_MgSi_2_O_8_) was observed in all samples, which indicated that all Ca(OH)_2_, MgSO_4_, MgSO_3_, Al_2_O_3_, CaCO_3_, and SiO_2_ in the raw materials had been chemically combined after calcination. Moreover, the peak intensities of anhydrite decreased with an increasing calcination temperature. When the calcination temperature was lower than 1250 °C, periclase (MgO), dicalcium silicate (Ca_2_SiO_4_), residual anhydrite (CaSO_4_), less mineral-phase magnesia alumina spinel (MgAl_2_O_4_), and merwinite (Ca_3_MgSi_2_O_8_) were produced in the SAC-MKPC clinker system, but ye’elimite was not produced. With an increase in the calcination temperature, ye’elimite appeared in the sample prepared at 1300 °C. The XRD patterns of SAC-MKPC clinker powders with calcination temperatures between 1300 and 1350 °C are shown in Figure 6 and Figure 7. The final clinker was mainly composed of ye’elimite ($C_{4}A_{3}\bar{S}$), periclase (MgO), and dicalcium silicate (C_2_S). A small amount of magnesia aluminum spinel (MgAl_2_O_4_) and akermanite (Ca_3_MgSi_2_O_8_) was observed in the calcination products. An abundance of periclase (MgO) and ye’elimite and a small amount of MgAl_2_O_4_ and Ca_3_MgSi_2_O_8_ were visible. MgAl_2_O_4_ was one of the mineral phases of the aluminum slag, and the residual anhydrite (CaSO_4_) had decomposed completely.

According to the XRD patterns of SAC-MKPC clinker powders with calcination temperatures between 1200 and 1350 °C under different raw material ratios, the desired mineral ($C_{4}A_{3}\bar{S}$, MgO) was produced. Meanwhile, aluminum spinel (MgAl_2_O_4_) and akermanite (Ca_3_MgSi_2_O_8_) were also found in the SAC-MKPC clinker system. Compared with the magnesium phosphate cementitious material, SAC-MKPC clinker can be prepared in a relatively low-temperature region.

### 3.2. Analysis of Hydration Products

To explore the possible mineral types of the SAC-MKPC hydration reaction, XRD tests of SAC-MKPC hydration were carried out. Figure 8 shows the XRD pattern of the SAC-MKPC pastes produced with industrial solid wastes at 1300 °C and cured for 28 d. The SAC-MKPC clinker was selected with a MgO theoretical content of 60% and the M/P was 5. The results show that the main minerals in the SAC-MKPC hydration reaction are K-struvite (MgKPO_4_·6H_2_O), ettringite (3CaO·Al_2_O_3_·3CaSO_4_·32H_2_O), and MgO, and unreacted ye’elimite ($C_{4}A_{3}\bar{S}$) and magnesia alumina spinel (MgAl_2_O_4_) were found in the XRD patterns. Hence, the goal of using coal gangue, carbide slag, magnesium desulfurization slag, and aluminum slag as raw materials to prepare the sulphoaluminate magnesium–potassium phosphate cementitious composite material was initially achieved.

The TG-DSC results of the SAC-MKPC pastes prepared with calcination temperatures between 1200 and 1350 °C and cured for 28 days are shown in Figure 9. The hydration products of SAC-MKPC include hydrated calcium aluminate sulfate (AFt-ettringite and AFm-monosulfate), MgKPO_4_·6H_2_O, aluminum glue, iron glue, and calcium silicate hydrate gel according to previous studies [30,31,32]. Meanwhile, unreacted ye’elimite, MgO, anhydrite, and KH_2_PO_4_ are also present. Hence, the thermal mass loss of the hydration products of SAC-MKPC can be divided into two main stages. The first stage’s temperature range is from 30 to 200 °C, and MgKPO_4_·6H_2_O, Aft, iron glue, and CaSO_4_·2H_2_O lose water. The second stage’s temperature range is from 30 to 200 °C, and unreacted KH_2_PO_4_ and aluminum glue lose water. The main reactions as follows:

First stage~
(4)$${\mathrm{MgKPO}}_{4}\cdot {6H}_{2}O\overset{60–200\;{}^{\circ}C}{\to }{\mathrm{MgKPO}}_{4}+{6H}_{2}O↑$$
(5)$$3\mathrm{CaO}\cdot {\mathrm{Al}}_{2}O_{3}\cdot {3\mathrm{CaSO}}_{4}\cdot {32H}_{2}O\overset{>90\;{}^{\circ}C}{\to }3\mathrm{CaO}\cdot {\mathrm{Al}}_{2}O_{3}\cdot {\mathrm{CaSO}}_{4}\cdot {12H}_{2}O+{20H}_{2}O↑$$
(6)$${\mathrm{CaSO}}_{4}\cdot {2H}_{2}O\overset{110–190\;{}^{\circ}C}{\to }{\mathrm{CaSO}}_{4}\cdot \frac{1}{2}H_{2}O+\frac{3}{2}H_{2}O↑$$
(7)$${\mathrm{Fe}}_{2}O_{3}\cdot {3H}_{2}O(\mathrm{gel})\overset{150–160\;{}^{\circ}C}{\to }{\mathrm{Fe}}_{2}O_{3}+{3H}_{2}O↑$$

Second stage~
(8)$${\mathrm{KH}}_{2}{\mathrm{PO}}_{4}\overset{>200\;{}^{\circ}C}{\to }{\mathrm{KPO}}_{3}+H_{2}O↑$$
(9)$${\mathrm{Al}}_{2}O_{3}\cdot {3H}_{2}O(\mathrm{gel})\overset{>300\;{}^{\circ}C}{\to }{\mathrm{Al}}_{2}O_{3}+{3H}_{2}O↑$$
(10)$${\mathrm{CaSO}}_{4}\cdot \frac{1}{2}H_{2}O\overset{>220\;{}^{\circ}C}{\to }{\mathrm{CaSO}}_{4}+\frac{1}{2}H_{2}O↑$$

As shown by the TG-DTG outputs of the SAC-MKPC hydration product, the hydration product starts to decompose at 60 °C, and the decomposition rate of the hydration product becomes faster as the temperature increases to 90~110 °C. Only one obvious instance of weight loss below 200 °C can be observed in the curves and can mainly be attributed to the decomposition of struvite and AFt. When the temperature increased to 230 °C, the hydration products of SAC-MKPC prepared by the T1200–1250 °C with MgO with a theoretical content of 40%, 50%, and 60% showed a small endothermic peak, and endothermic peaks were not observed for the other samples in the temperature range. When the temperature exceeded 400 °C, the mass loss of the hydration product remained stable. 

Figure 9a shows the TG-DTG results of the SAC-MKPC pastes prepared with a calcination temperature of 1200 °C and cured for 28 days. The endothermic peaks of the hydration products appeared at 60–200 °C, which indicates the main phase decomposition of MgKPO_4_·6H_2_O and CaSO_4_·2H_2_O. Another obvious instance of weight loss at 230–260 °C can be observed in the curves and can mainly be attributed to the decomposition of unreacted KH_2_PO_4_. Figure 9b shows the TG-DTG results of the SAC-MKPC pastes prepared with a calcination temperature of 1250 °C and cured for 28 days. The endothermic peak of the hydration product was similar to that in Figure 9b at 60–200 °C and at 230–260 °C, which indicates the main phase decomposition of MgKPO_4_·6H_2_O, CaSO_4_·2H_2_O, AFt, and unreacted KH_2_PO_4_. Compared with Figure 9a,b, the endothermic peaks of the hydration products that appeared at 60–200 °C were similar to those in Figure 9c,d. However, there was no obvious endothermic peak above 200 °C, which indicated that unreacted KH_2_PO_4_ was nonexistent in the hydration products of SAC-MKPC. 

Table 5 summarizes the mass loss (WL_30–200 °C_ and WL_200–700 °C_) and the most intense temperature peak of mass loss (TPWL) in the different temperature ranges. When the calcination temperature was 1200–1250 °C and the MgO theoretical content was 40–50%, the most intense TPWL of the SAC-MKPC hydration products was low, at ~100 °C. When the calcination temperature was 1300–1350 °C with the same MgO theoretical content, the TPWL of the SAC-MKPC hydration products increased slightly, and the TPWL exceeded 100 °C and extended to 104 °C. The TPWL reached 110 °C with an increase in the theoretical MgO content at the same calcination temperature because more MgKPO_4_·6H_2_O formed in the hydration product.

Table 5 shows that all samples had an increase in mass loss with increasing theoretical MgO values at 30–200 °C, but all samples had a decrease in mass loss with increasing theoretical MgO values at 200–700 °C. At 30–200 °C, the MgO theoretical content was 60–80%, the largest mass loss of the SAC-MKPC hydration products occurred when the clinker was prepared at 1300 °C, followed by 1350 °C and 1200 °C, and the smallest mass loss occurred at 1250 °C. Therefore, the sample in which the MgO theoretical content was 60–80% at 1300 °C showed the highest AFt and MgKPO_4_·6H_2_O content compared with that in the other three samples. However, the mass loss of the SAC-MKPC hydration products when the clinker was prepared at 1200 °C was higher than that prepared at 1250 °C. Combined with the analysis of the mineral composition of the SAC-MKPC clinker in Section 3.1, the anhydrate content decreased with increasing temperature, and its content was highest, which was observed in the sample at 1200 °C, and the AFt phase was not generated by 3CaO·3Al_2_O_3_·CaSO_4_ hydration; hence, the hydration product phase at 1250 °C was lower. At 200–700 °C, the largest mass loss of the SAC-MKPC hydration product occurred at 1200 °C, followed by 1250 °C and 1300 °C, and the smallest mass loss occurred at 1350 °C. The main mass loss was caused by the decomposition of unreacted KH_2_PO_4_ in the same temperature range.

### 3.3. SEM-EDS Analysis

In the study of the microstructure of the SAC and MKPC hydration product, the calcium sulfoaluminate hydrate showed many structural forms, including fine needles, thick needles, and tubular, cylindrical, and hexagonal shapes. The aluminum glue (Al_2_O_3_·3H_2_O(gel)) was pom-pom- or needle-shaped [33], and the MgKPO_4_·6H_2_O had a block or plate crystal structure [34,35].

Figure 10 shows the SEM micrographs of the SAC-MKPC and the EDS results (curing age: 28 days). Large crystals in a prism-like shape, a cylindrical shape, and a pom-pom shape were found in the paste (Figure 10). Energy spectrum analysis for four different surface areas (A, B, C, D) was invested, A area is a prism-like shape, B area is a skeleton structure, C area is a block shape, D area is a pom-pom shape. Elemental distribution in selected areas is given in Table 6. The main elements (Ca, S, Al, Fe, and O) were visible in area A. The molar ratios (m.r.) of Ca:(Al+Fe):S were 4.74:1:3.3, which is close to 4:1:3, implying the existence of AFt [32]. According to the EDS results of area A and the XRD results mentioned above, these crystals with a regular columnar structure were proven to be AFt. Area B showed a solid block structure, which was covered with a thin plate crystal. According to the EDS results for area B, the main elements (Mg, K, P, and O) were visible in area B. This indicates that the molar ratio of P:K:Mg:O was 1.9:1.0:(1 + 8.72):23.55, which is close to the theoretical value (1:1:1:10) of the molar ratios of the MgO skeleton structure covered with MgKPO_4_·6H_2_O crystals. According to the EDS results for area C, the main elements were Mg, P, K, and Ca in area C, and the molar ratio (m.r.) of P:K:Mg was 1.26:1:0.92, which is close to the theoretical value (1:1:1) of the molar ratios of K-struvite (MgKPO_4_·6H_2_O). For area D, the surface of the bulk was covered by amorphous material that had a complicated element composition, which included Ca, Al, P, and a small amount of elemental Fe, S, and Si. This was mostly a cementation product of MgKPO_4_·6H_2_O, 3CaO·Al_2_O_3_·3CaSO_4_·32H_2_O, aluminum glue, iron glue, silica gel, calcium phosphate, and other minerals. The bulk could be glued together through the amorphous material to form a relatively dense structure.

### 3.4. Mechanical Properties of the SAC-MKPC Paste

The influence of the main-phase theoretical composition and the calcination temperature on the mechanical properties of the SAC-MKPC paste under different preparation conditions was investigated. The results are shown in Figure 11 and Figure 12.

Figure 11 shows the setting time of the different SAC-MKPC samples from the VICAT measurements. When the theoretical content of MgO remained unchanged, the setting time increased initially and then decreased with an increase in the calcination temperature. Combined with the analysis of the mineral composition of the SAC-MKPC clinker in Section 3.1, when the calcination temperature was 1200 °C, the main mineral phases of the SAC-MKPC clinker were MgO, CaSO_4_, a small amount of Ca_2_SiO_4_, and MgAl_2_O_4_ and Ca_3_MgSi_2_O_8_ impurities. The MgO of the SAC-MKPC clinker was light-burned magnesia, which has a higher activity and reacts rapidly with KH_2_PO_4_; hence, the setting time was short. As the calcination temperature continued to increase, the main mineral phases in the clinker transformed to MgO, 3CaO·3Al_2_O_3_·CaSO_4_, and Ca_2_SiO_4_. The MgAl_2_O_4_, Ca_3_MgSi_2_O_8_, and CaSO_4_ phases decreased gradually and disappeared at 1300 °C. The MgO phase of the SAC-MKPC clinker was dead-burnt magnesia, which had a low reactivity and increased the setting time of the SAC-MKPC cementitious composite material under the calcination conditions. When the calcination temperature increased to 1350 °C, the main mineral phases were MgO, Ca_2_SiO_4_, and 3CaO·3Al_2_O_3_·CaSO_4_ phases, and the total content of the main mineral phases increased slightly. Phases of MgAl_2_O_4_ and Ca_3_MgSi_2_O_8_ impurities of a lower content had little effect on the setting time. The setting time was similar to that of the pure SAC-MKPC slurry that was prepared for the calcination temperature at 1300 °C.

When the calcination temperature remained unchanged, the setting time increased initially and then decreased with the increase in the theoretical content of MgO. When the theoretical content of MgO reached the maximum, the setting time was the shortest. When the theoretical content of MgO in the SAC-MKPC clinker exceeded 60%, the contact probability was higher between the MgO and the KH_2_PO_4_ dissolved in water, which resulted in a rapid acid–base reaction and the SAC-MKPC had the shortest setting time. When the theoretical content of MgO in the clinker was 40–60%, the setting time increased gradually with an increase in the theoretical MgO content. The clinker contained a certain amount of 3CaO·3Al_2_O_3_·CaSO_4_ and residual Al_2_O_3_ and MgO. Al_2_O_3_ and MgO reacted with KH_2_PO_4_ in water, which increased the setting time gradually. The setting time of the SAC-MKPC composite material was closer to the hydration reaction time of 3CaO·3Al_2_O_3_·CaSO_4_.

The compressive strength of the SAC-MKPC pastes prepared with different theoretical MgO contents at the same calcination temperature for various curing ages is presented in Figure 12. The compressive strength of samples with a theoretical MgO content of 40% and 50% was low at curing ages of 2 h, 1 d, 3 d, 7 d, and 28 d, respectively, and it did not reach the relevant national standard. The early (from 2 h to 3 d) and later (28 d) compressive strengths of the SAC-MKPC increased significantly with an increase in the MgO content. The content of theoretical MgO was 70% and 80%. SAC-MKPC had a higher early strength and late strength. The compressive strength of the sample cured for 2 h exceeded 30 MPa and the compressive strength of the sample cured for 1 d reached 80% of the 28 d compressive strength and later stabilized.

Figure 12 shows the effects of the calcination temperature on the compressive strengths of the SAC-MKPC with the same raw material composites. The compressive strength of the SAC-MKPC samples was low when the calcination temperature was 1200 °C and the theoretical MgO content of the SAC-MKPC clinker was less than 60%. However, when the calcination temperature was unchanged, the theoretical MgO content was 70% and 80% and the 2-h compressive strength and late strength were higher than for other theoretical MgO contents. The 3-d compressive strength was close to the 28-d compressive strength, and the later strength was relatively stable.

### 3.5. Water Resistance of the SAC-MKPC Paste

To obtain the water resistance of the SAC-MKPC paste, the effects of M/P, ingredients, and curing conditions on the water resistance of SAC-MKPC were validated. According to the mechanical properties of the SAC-MKPC paste (Section 3.4) and using the clinker prepared at a calcination temperature of 1300 °C and with a theoretical MgO of 60%, 70%, 80%, and 100% as the foundational material, the M/P was set to 3/1, 5/1, 7/1, and 9/1 using the above-described method (Section 2.2) to achieve the different water requirements. The compressive strength was obtained at different curing ages and under different curing conditions (air and water). To present the water resistance of the SAC-MKPC paste clearly, the strength retention rate [35,36,37] was defined as:(11)K=f/F
where K denotes the compressive strength retention rate of the sample, *f* denotes the compressive strength of the sample at 28 days (MPa) in water, and F denotes the compressive strength of the sample at 28 days (MPa) after air curing. The compressive strength of the sample at 28 days (MPa) for different curing styles and strength retention were also shown in Table 7.

Figure 13 shows the compressive strength retention rate of SAC-MKPC with different MgO contents and the M/P for two different curing schedules (air and water). When the M/P was 5, 7, and 9, the theoretical MgO content increased from 60% to 70%, and the compressive strength retention rate decreased slowly. When the theoretical MgO content exceeded 70%, the compression strength retention rate gradually increased; that is, the water resistance of the SAC-MKPC paste decreased with an increase in the theoretical MgO content. When the M/P was 3, the compressive retention rate of the SAC-MKPC showed a significant decrease with an increase in the theoretical MgO content. Hence, the water resistance of the SAC-MKPC worsened with an increase in the MgO content when the M/P was constant. When the theoretical MgO content was 60–70%, the water resistance of the SAC-MKPC improved.

Figure 13 also shows that the compressive strength retention rate of the SAC-MKPC decreased with an increase in the M/P when the theoretical MgO content was constant. When the theoretical MgO content remained unchanged, the water resistance of the SAC-MKPC increased first and then decreased with the increase in the M/P. When the M/P was 7, the compressive strength retention rate was the largest; that is, the SAC-MKPC had the highest water resistance. According to the above result, when the M/P was 5, the theoretical MgO contents were 60% and 70%; when the M/P was 7, the theoretical MgO content was 60% and 70%; and when the M/P was 9, the theoretical MgO content was 60%. For a compressive strength retention rate K greater than 1, the test blocks that were prepared by SAC-MKPC did not experience a decrease in strength during water curing, and the compressive strength increased slightly. The main reasons for the strength decrease and the reverse enhancement in the SAC-MKPC test block cured in a water environment were that the M/P was too high, the proportion of sulfoaluminate in the SAC-MKPC system was high, no unreacted KH_2_PO_4_ existed in the hydration product, the SAC-MKPC block did not form pores that were dissolved by KH_2_PO_4_ during water curing, the dense structure was not destroyed, and the strength did not decrease. Although the hydration product MgKPO_4_·6H_2_O was dissolved partially in the acidic water environment, micropores formed by the dissolution of the unreacted calcium sulfoaluminate could be hydrated to generate ettringite, and they filled the micropores. The compactness of the inner structure improved, and the compressive strength retention rate increased [38,39,40,41]. The SAC-MKPC system exhibited better mechanical properties. Hence, the SAC-MKPC cementitious composite material was prepared at a low calcination temperature.

### 3.6. Reaction Pathways of Mineral Formation

Based on the performance analysis of the SAC-MKPC prepared from industrial solid wastes, the main formation mechanism of SAC-MKPC can be proposed.

The mineral phase of carbide slag decomposed into calcium oxide, water, and carbon dioxide.
(12)$${\mathrm{Ca}(\mathrm{OH})}_{2}\to \mathrm{CaO}+H_{2}O$$
(13)$${\mathrm{CaCO}}_{3}\to \mathrm{CaO}+{\mathrm{CO}}_{2}$$

Magnesium sulfite hexahydrate of the magnesium desulfurization slag decomposed into magnesium oxide, sulfur dioxide, and oxygen.
(14)$${\mathrm{MgSO}}_{3}\to \mathrm{MgO}+{\mathrm{SO}}_{2}$$
(15)$${2\mathrm{MgSO}}_{4}\to 2\mathrm{MgO}+2{\mathrm{SO}}_{2}+O_{2}$$

A more stable calcium sulfate phase formed from sulfur dioxide, oxygen, and calcium oxide.
(16)$$2\mathrm{CaO}+2{\mathrm{SO}}_{2}+O_{2}\to {2\mathrm{CaSO}}_{4}$$

Calcium oxide, aluminium oxide, and calcium sulfate transformed into the ye’elimite phase. Magnesium oxide became dead-burnt magnesium oxide.
(17)$$3\mathrm{CaO}+{3\mathrm{Al}}_{2}O_{3}+{\mathrm{CaSO}}_{4}\to 3\mathrm{CaO}\cdot {3\mathrm{Al}}_{2}O_{3}\cdot {\mathrm{CaSO}}_{4}$$
(18)MgO→dead-burnt MgO

MKPC was generated based on an acid-neutralization reaction. Equation (17) shows that the MKPC generation process is a high-speed reaction compared with SAC hydration. Therefore, MKPC was an early stage reaction product, which is critical to the early stage compressive strength.
(19)$$\mathrm{MgO}+{\mathrm{KH}}_{2}{\mathrm{PO}}_{4}+{5H}_{2}O\to {\mathrm{MgKPO}}_{4}\cdot {6H}_{2}O$$
$$3\mathrm{CaO}\cdot {3\mathrm{Al}}_{2}O_{3}\cdot {\mathrm{CaSO}}_{4}+{2(\mathrm{CaSO}}_{4}\cdot {2H}_{2}O)+{34H}_{2}O\to$$
(20)$$3\mathrm{CaO}\cdot {\mathrm{Al}}_{2}O_{3}\cdot {3\mathrm{CaSO}}_{4}\cdot {32H}_{2}O+{2(\mathrm{Al}}_{2}O_{3}\cdot {3H}_{2}O)(\mathrm{gel})$$

In the SAC-MKPC system, the K-struvite crystal formed first and the AFt formed later. The K-struvite crystal changed gradually into a large bulk phase and the K-struvite surface was covered with AFt crystals. The two main mineral phases, k-struvite and AFt, were linked, and therefore they formed a dense structure [42,43,44].

## 4. Conclusions

We conducted experimental research on the SAC-MKPC cementitious composite material that was produced from aluminum slag, carbide slag, coal gangue, and magnesium desulfurization slag, with a focus on the theoretical feasibility of preparing SAC-MKPC clinker from industrial solid wastes and the main formation mechanism. The mechanical strength of the hydration product, the phases in the clinker, and the thermogravimetry, microstructure, and water resistance of the SAC-MKPC paste were researched in detail. According to the experimental results, our conclusions are as follows:(1)The expected main mineral phases (MgO, ye’elimite, and Ca_2_SiO_4_) formed in the SAC-MKPC clinker, and the calcination temperature of the main mineral phases formed was between 1250 °C and 1350 °C.(2)The SAC-MKPC had better strength behavior when the calcination temperature was ~1300 °C, the theoretical MgO content was 60–70%, and the M/P was 5 and 7. The best compressive strength reached 35.2, 70.9, 84.1, 87.7, and 101.6 MPa at 2 h, 1 d, 3 d, 7 d, and 28 d of hydration, respectively.(3)The XRD analysis of the hydration products of the SAC-MKPC composite indicated that K-struvite and ettringite coexisted. The SEM micrographs also showed that the mutual adhesion of the FAt and K-struvite crystals led to the formation of a very dense structure. The dense structure provided the SAC-MKPC with excellent water resistance. This novel preparation method could use industrial solid wastes as raw materials to prepare SAC-MKPC cementitious composite materials of high value.

## Figures and Tables

**Figure 1 materials-14-07340-f001:**
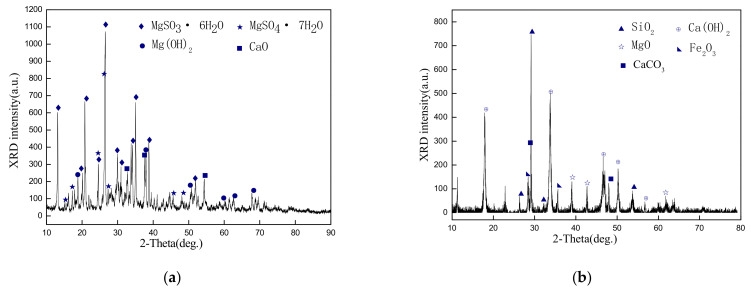
XRD patterns of raw materials: (**a**) magnesium desulfurization slag; (**b**) carbide slag; (**c**) aluminum slag; (**d**) coal gangue.

**Figure 2 materials-14-07340-f002:**
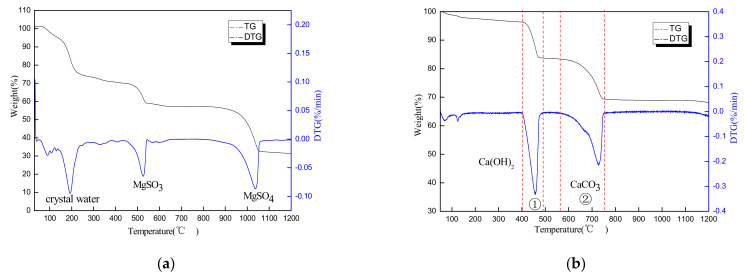
TG−DTG outputs of raw materials: (**a**) magnesium desulfurization slag; (**b**) carbide slag.

**Figure 3 materials-14-07340-f003:**
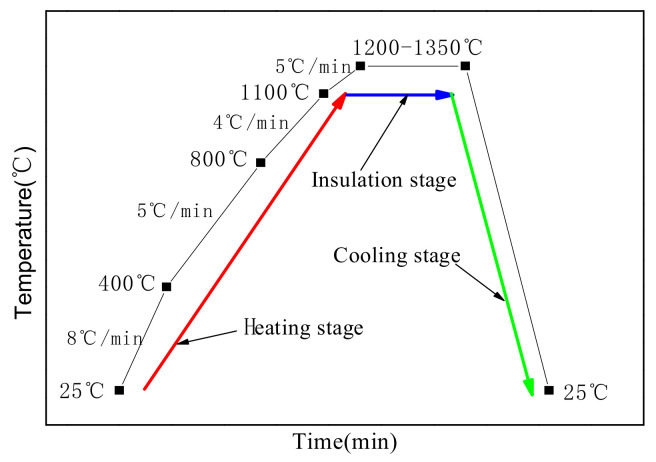
Calcination temperature increase control curve.

**Figure 4 materials-14-07340-f004:**
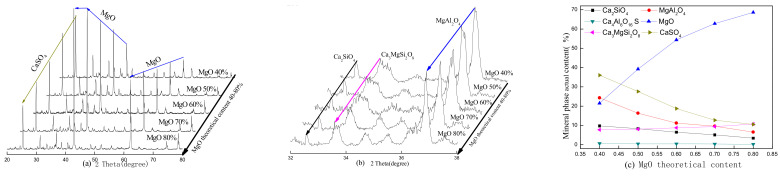
XRD patterns of the SAC-MKPC clinker with MgO theoretical content at 1200 °C: (**a**) main phase; (**b**) minor phase; and (**c**) mineral phase actual content.

**Figure 5 materials-14-07340-f005:**
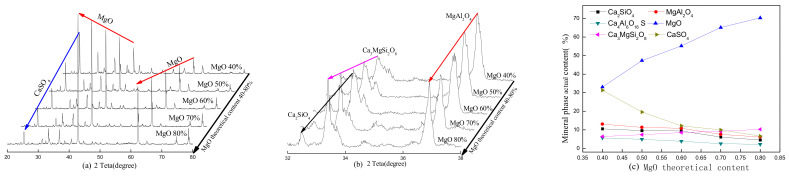
XRD patterns of the SAC-MKPC clinker with MgO theoretical content at 1250 °C: (**a**) main phase; (**b**) minor phase; and (**c**) mineral phase actual content.

**Figure 6 materials-14-07340-f006:**
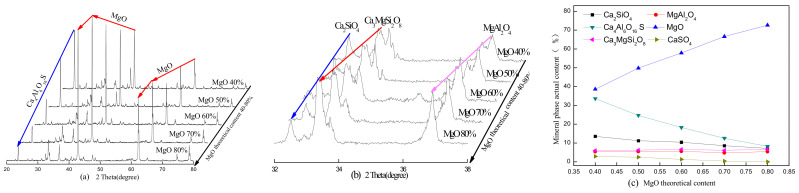
XRD patterns of the SAC-MKPC clinker with MgO theoretical content at 1300 °C: (**a**) main phase; (**b**) minor phase; and (**c**) mineral phase actual content.

**Figure 7 materials-14-07340-f007:**
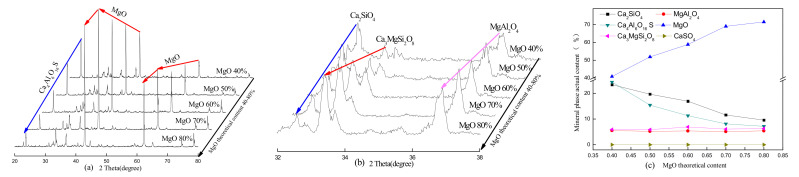
XRD patterns of the SAC-MKPC clinker with MgO theoretical content at 1350 °C: (**a**) main phase; (**b**) minor phase; and (**c**) mineral phase actual content.

**Figure 8 materials-14-07340-f008:**
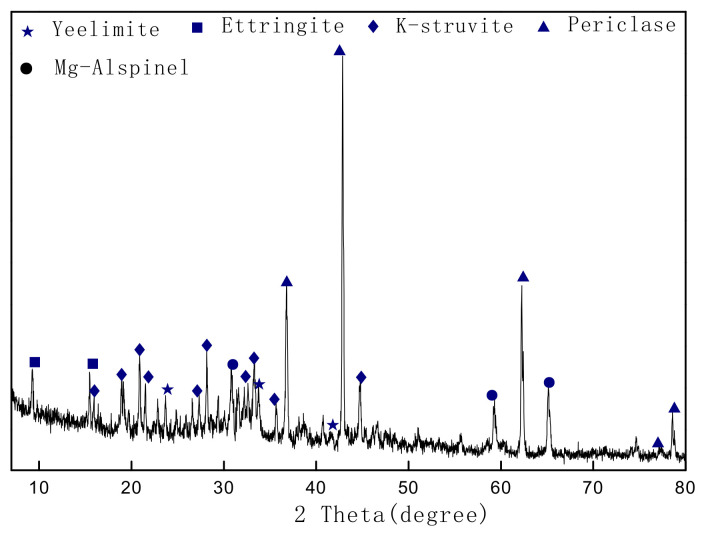
XRD pattern of SAC-MKPC paste after 28 days of curing.

**Figure 9 materials-14-07340-f009:**
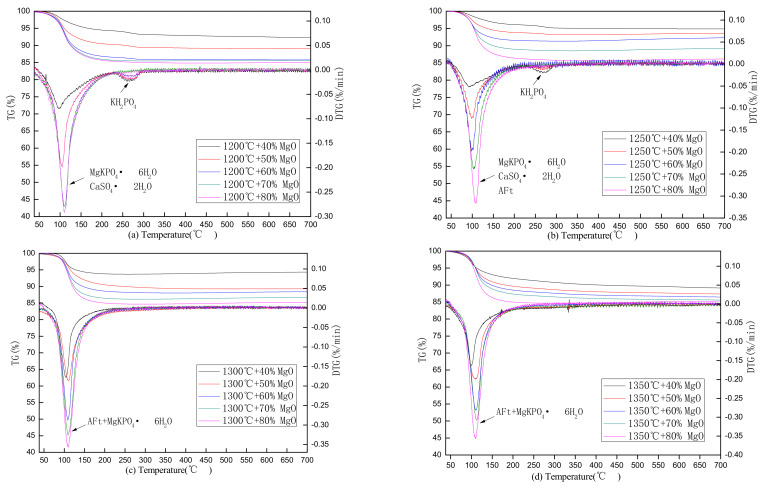
TG-DTG outputs of the SAC-MKPC hydration product at 28 days: (**a**) 1200 °C; (**b**) 1250 °C; (**c**) 1300 °C; (**d**) 1350 °C.

**Figure 10 materials-14-07340-f010:**
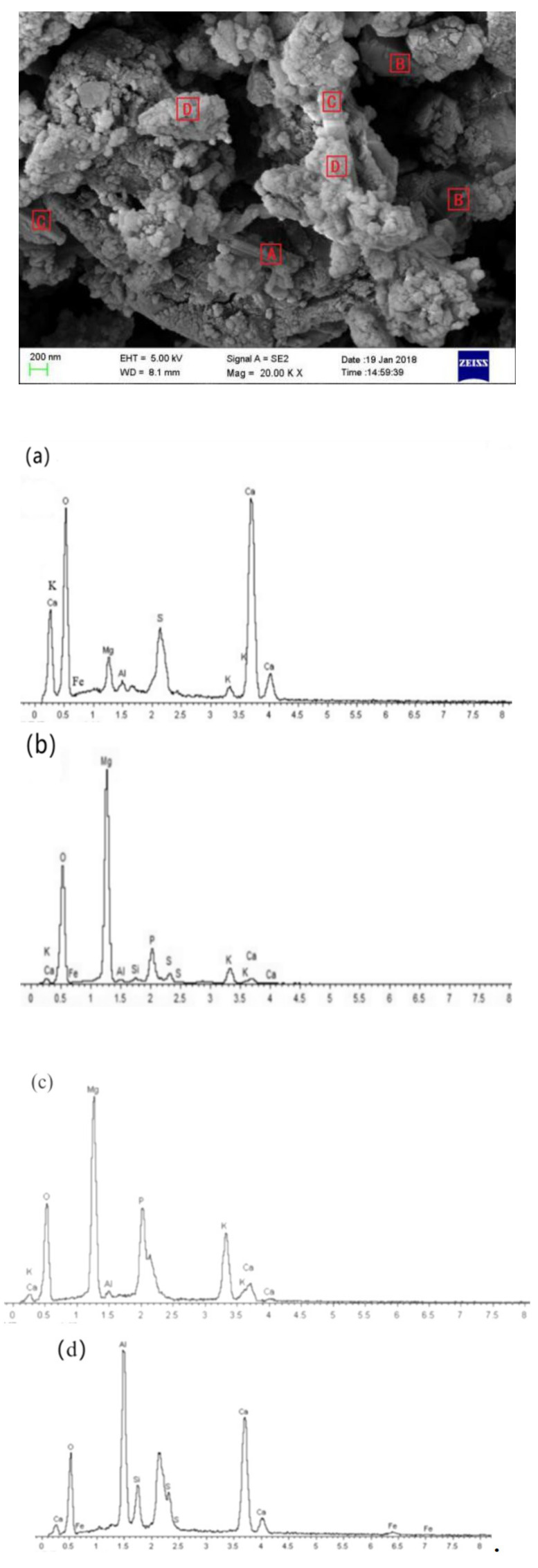
SEM micrographs of SAC-MKPC and EDS spectra (curing age: 28 days) (**a**. **A** area, **b**. **B** area, **c**. **C** area, **d**. **D** area).

**Figure 11 materials-14-07340-f011:**
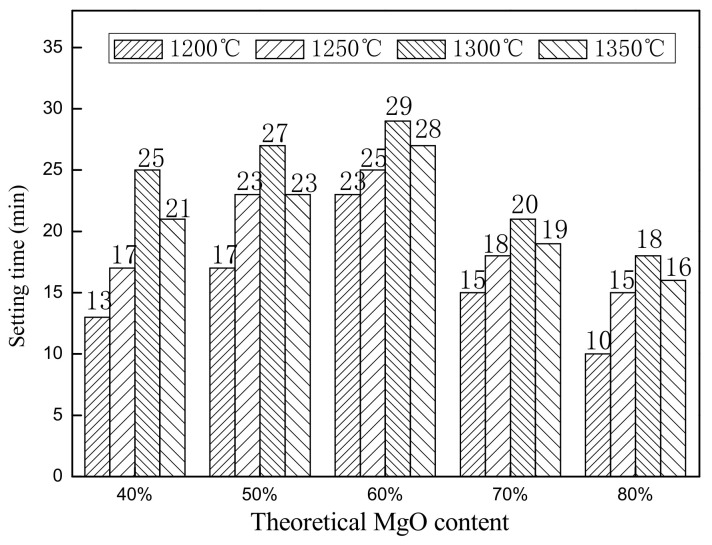
SAC-MKPC sample setting time.

**Figure 12 materials-14-07340-f012:**
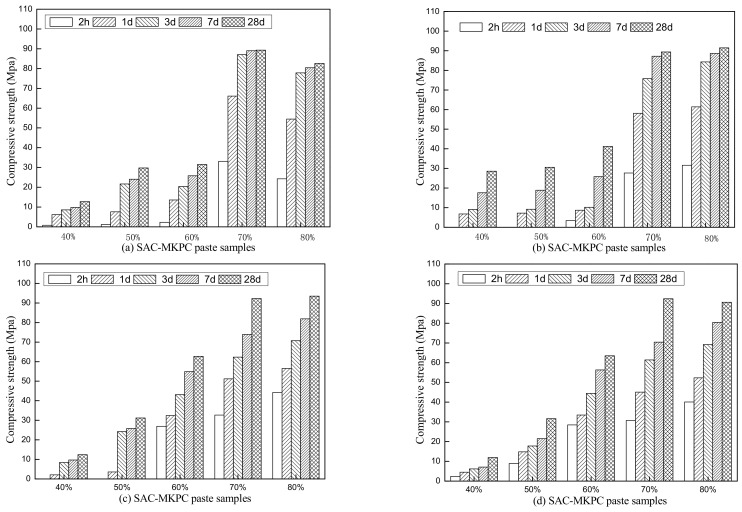
Compressive strength of SAC-MKPC samples at (**a**) 1200 °C, (**b**) 1250 °C, (**c**) 1300 °C, and (**d**) 1350 °C.

**Figure 13 materials-14-07340-f013:**
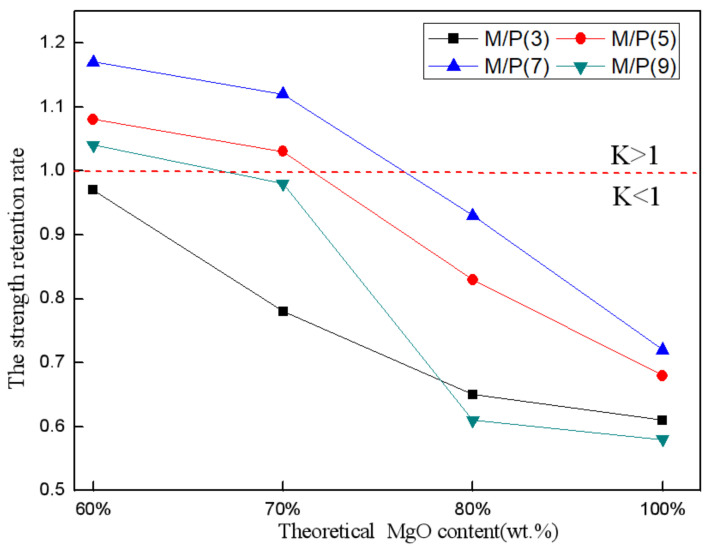
Water resistance of the SAC-MKPC.

**Table 1 materials-14-07340-t001:** Main pieces of experimental equipment.

Process	Experimental Instruments	Manufacturer/Model	Country
Raw Material Preparation Equipment	Electronic balance	Shanghai Yueping Scientific Instrument Co., Ltd.; FA2004B	China
	Disc refiner	Nanjing University Instrument Plant; QM-3SP04	China
	Sample pulverizer	Shanghai Shuli Yiqi Yibiao Co., Ltd.; GJ100-1A	China
	Hot air oven	Shandong Luda Experiment Instrument Co., Ltd.; DHG-9053	China
Clinker Calcination and Molding Equipment	Standard sieve	Shandong Luda Experiment Instrument Co., Ltd.; 200 mesh	China
	Box resistance furnace	Hennan Jianxi Experiment Instrument Co., Ltd.; KSL-1600X	China
	Steel mold	Shandong Luda Experiment Instrument Co., Ltd.; 20 mm × 20 mm × 20 mm	China
	Cement shaker	Shandong Luda Experiment Instrument Co., Ltd.; 60 times/min	China
	Standard curing box	Shandong Luda Experiment Instrument Co., Ltd.; YH40B	China
	Cement mortar vibration table	Shandong Luda Experiment Instrument Co., Ltd.; 170 mm × 110 mm × 300 mm	China
Analysis Equipment	Automatic pressure measurement testing machine	Shandong Luda Experiment Instrument Co., Ltd.; DYH-300 B	China
	Automatic setting time tester	Jian Yan Hua Ce (Hangzhou) Science & Technology Co., Ltd.	China
	SEM-EDS	Fei Electron Microscope Co., Ltd.; Quanta200;	Netherlands
	X-ray diffraction	Europe Italy Boris Pastemak Co., Ltd.; Europe	Germany
	X-ray fluorescence	USA Thermal Scientific Co., Ltd.; D8-Advance	America
	TG-DTG	NETZSCH STA 409 PC/PG Thermal Analyzer	

**Table 2 materials-14-07340-t002:** Chemical composition of the raw materials (wt%).

	MgO	Al_2_O_3_	SiO_2_	SO_3_	CaO	Fe_2_O_3_	TiO_2_	R_2_O ^a^	LOI ^b^	SAM ^c^
Carbide Slag	0.34	1.33	1.41	1.24	75.05	0.25	0.03	0.21	20.14	KH_2_PO_4_ (99%, Aladdin)
Aluminum Slag	4.87	70.79	9.65	0.41	1.95	4.11	0.49	3.51	4.22	
Coal Gangue	2.55	20.16	61.62	1.95	2.27	3.28	1.24	0.71	6.22	
MDS	31.46	1.21	1.03	58.15	1.26	0.31	0.12	0.23	6.23	

^a^ Alkaline oxide (K_2_O, Na_2_O). ^b^ Loss on ignition at 950 °C. ^c^ Secondary added materials.

**Table 3 materials-14-07340-t003:** Compositions of raw mixes in the experiment with different calcination temperatures.

Sample	Coal Gangue/g	Aluminum Slag/g	MDS/g	Carbide Slag/g	Calcination Temperature/°C	Holding Time/Min	Temperature Interval/°C	MgO Theoretical Content/wt%
A	1.22	14.33	63.74	20.71	1200–1350	30 min	50	40%
B	0.75	8.63	76.73	13.89	1200–1350	30 min	50	50%
C	0.43	5.09	84.86	9.62	1200–1350	30 min	50	60%
D	0.23	2.71	90.43	6.63	1200–1350	30 min	50	70%
E	0.15	1.67	93.07	5.11	1200–1350	30 min	50	80%
F	0	0	100	0	1200–1350	30 min	50	100%

**Table 4 materials-14-07340-t004:** Mix proportions for the secondary mixing (g/100 g clinker).

Sample	Al_2_O_3_	Fe_2_O_3_	CaO	SO_3_	MgO	K_2_O	SiO_2_	Water	KDP	Gypsum
1200–40%	12.19	1.18	31.23	24.07	25.06	0.19	5.69	29.4	17.03	3.73
1200–50%	8.65	1.22	25.29	18.72	40.87	0.11	4.85	29.6	27.79	2.94
1200–60%	5.32	1.36	20.08	14.22	54.36	0.06	4.35	29.7	36.96	2.26
1200–70%	3.05	1.61	16.08	12.48	62.68	0.03	3.88	29.8	42.62	1.85
1200–80%	2.05	1.64	13.58	10.05	68.03	0.06	4.42	29.9	46.23	1.59
1250–40%	13.37	1.18	29.83	25.24	24.81	0.13	5.01	29.4	16.87	3.73
1250–50%	11.98	1.38	24.95	15.77	40.42	0.14	4.96	29.6	27.48	2.95
1250–60%	8.78	1.59	20.43	10.13	52.84	0.08	5.72	29.7	35.93	2.33
1250–70%	3.81	1.63	15.25	7.76	66.45	0.04	4.85	29.8	45.18	1.66
1250–80%	2.82	1.76	12.55	4.45	73.54	0.08	4.61	29.9	50.01	1.31
1300–40%	25.18	1.26	32.13	8.89	24.66	0.21	6.93	29.4	16.77	3.73
1300–50%	19.85	1.54	28.81	6.73	34.69	0.17	7.6	29.6	23.59	3.23
1300–60%	14.24	2.39	22.96	4.73	46.33	0.23	8.46	29.7	27.78	2.65
1300–70%	7.75	2.48	20.79	5.78	54.56	0.07	8.26	29.8	37.11	2.25
1300–80%	4.48	2.55	16.52	1.81	66.14	0.05	8.24	29.9	44.98	1.68
1350–40%	26.13	1.42	35.00	4.76	23.85	0.13	8.17	29.4	16.22	3.78
1350–50%	19.24	1.63	30.49	3.26	36.01	0.08	8.81	29.6	24.48	3.17
1350–60%	14.46	1.72	23.82	3.89	47.03	0.13	8.58	29.7	31.97	2.63
1350–70%	7.17	2.18	19.19	3.84	58.57	0.07	8.68	29.8	41.19	2.05
1350–80%	7.99	2.97	15.48	2.14	62.67	0.16	8.04	29.9	42.86	1.84

**Table 5 materials-14-07340-t005:** TG results of the SAC-MKPC hydration product at 28 days.

Sample	1200–40%	1200–50%	1200–60%	1200–70%	1200–80%
TPWL, °C	98	102.5	106	107.5	108
WL30–200 °C, wt.%	5.51	9.48	13.16	14.04	14.31
WL200–700 °C, wt.%	2.24	1.45	1.08	0.87	0.68
Sample	1250–40%	1250–50%	1250–60%	1250–70%	1250–80%
TPWL,°C	92.5	97.5	101.5	105	108
WL30–200 °C, wt.%	4.87	7.16	8.45	11.02	13.59
WL200–700 °C, wt.%	2.15	1.09	0.75	0.45	0.25
Sample	1300–40%	1300–50%	1300–60%	1300–70%	1300–80%
TPWL,°C	103	105	106.5	107.5	108
WL30–200 °C, wt.%	6.19	9.49	11.41	13.55	14.99
WL200–700 °C, wt.%	1.54	1.24	0.86	0.53	0.46
Sample	1350–40%	1350–50%	1350–60%	1350–70%	1350–80%
TPWL, °C	102	104	106	107	108
WL30–200 °C, wt.%	8.02	10.33	11.18	12.63	14.61
WL200–700 °C, wt.%	1.43	1.13	0.74	0.58	0.41

Note: TPWL, temperature peak of mass loss, °C; WL_30–200 °C_, 30–200 °C mass loss percentage; WL_200–700 °C_, 200–700 °C mass loss percentage.

**Table 6 materials-14-07340-t006:** Elemental distribution in selected areas.

Element	A		B		C		D	
	at%	m.r.	at%	m.r.	at%	m.r.	at%	m.r.
C	1.51	2.88	-	-	-	-	0.58	1.10
O	43.83	62.68	49.17	61.94	43.34	3.26	42.23	60.02
Al	3.28	2.78	0.48	0.36	0.95	0.76	18.88	15.91
Si	-	-	0.54	0.39	-	-	2.24	1.74
S	17.32	12.36	1.99	1.25	-	-	2.03	1.44
Ca	31.09	17.75	1.61	0.81	4.47	2.42	29.01	16.49
Fe	2.34	0.96	0.19	0.07	-	-	1.97	0.81
Mg	0.63	0.59	33.25	27.57	19.61	15.07	-	-
P	-	-	7.69	5.00	13.41	11.99	3.09	2.51
K	-	-	5.09	2.63	18.22	11.00	-	-

**Table 7 materials-14-07340-t007:** Strength retention rate under different curing conditions.

M/P	Theoretical MgO Content	Air Curing for 28 d (F/MPa)	Water Curing for 28 d (*f*/MPa)	Strength Retention Rate (K)
3/1	100%	53.9	32.9	0.61
	80%	67.5	43.8	0.65
	70%	64.6	50.4	0.78
	60%	48.3	46.9	0.97
5/1	100%	81.5	55.4	0.68
	80%	91.4	76.5	0.83
	70%	89.5	92.2	1.03
	60%	53.5	57.8	1.08
7/1	100%	83.5	60.1	0.72
	80%	92.6	86.1	0.93
	70%	90.7	101.6	1.12
	60%	55.2	64.6	1.17
9/1	100%	42.2	24.5	0.58
	80%	31.5	19.2	0.61
	70%	25.2	24.7	0.98
	60%	12.4	12.9	1.04
SAC	-	68.7	76.95	1.12

## Data Availability

The data presented in this study are available on request from the corresponding author.

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
