# Peer review of "Preparation and Properties of a Sulphoaluminate Magnesium-Potassium Phosphate Green Cementitious Composite Material from Industrial Solid Wastes"

_materials, 2021, doi:10.3390/ma14237340_

Round 1

Reviewer 1 Report

Best regards

Author Response

Revisions and Responses

General Responses to the Editor

Dear Professor.

Thank you and the reviewers so much for your efforts and precious comments on our manuscript.

After receiving your letter, our working group made a thorough discussion about all the comments and advice. Then, a careful revision was implemented and the details were listed in the following part.

We hope the revised version could meet the publication requirements in Materials. If you have any other concerns, we would like to give further revisions or responses as soon as we can.

Best regards and many thanks!

Changzai Ren and all the other authors

Responses to reviewer #1:

Reviewer #1: I read the above manuscript with much interest, as it deals with an important issue related to new materials, also addressing some environmental aspects with implications on the management of solid waste. The manuscript contains some new data and describes processes occurring during the preparation of a cementitious material. The text contains paragraphs which are well-written and several others not well-written at all in terms of grammar and syntax, as well as in terms of organization.
General Response:Thank the reviewer very much for your efforts on our paper and for your approval to the work! Following your comments, we have thoroughly revised the manuscript and improved the English expressions. Thank you sincerely for giving us the opportunity to improve our paper to a new level. If you have any other concern, we would like to give further responses as soon as we can.

Specific comments:

  1. Under the chapter 2.1 Materials, the authors provide details about methodology (XRF, XRD, quantification of phases, etc.) and much later in chapter 2.3, they provide the details and operating conditions for these instruments. I expected that the latter should precede the materials and every analytical results. Table 1 contains data normalized to 100% and I wonder why. Why not the raw results? In this way the readers would be able to validate the accuracy and the reliability of the data. The legend of Table 3 refers that the units are g/100 g clinker, however if we consider hydrous basis,the provided values far exceed 100 g or if we consider a dry basis they are again different than 100 g.What are these values?
    1.Response:Thank you very much for your advice. Considering the reviewer’s suggestion, we have revised the Section 2, the chapter 2.3 has been revised as the first paragraph in Section 2. The normalized data of raw materials should not be modified in manuscript. As the normalized data was used to calculate the content of the main phase in clinker system, the SAC-MKPC cementitious composite material clinker modulus is controlled at an alkalinity modulus Cm, an alumina– sulfur ratio P and an alumina–silica ratio N and MgO content. The CaO, Al2O3, Fe2O3, SiO2 were all the normalized data.

Alkalinity modulus (Cm):   (1)

Alumina-sulfur ration(P):                            (2)

Alumina-silica ration(N):                                     (3)

The data normalized to 100% was found in many research, such as the following reference:

1.Yu,J.; Qian,J.S.;Wang,F.; Li,Z.; Jia,X.W.;Preparation and properties of a magnesium phosphate cement with dolomite,Cement and Concrete Research,2020,138,106235.

Thank you very much for your advice and we are sorry for the ambiguous expression“g/100g clinker”. For the table 3, the every SAC-MKPC clinker sample was calcined at 1200-1300℃, they are not hydrous basis. And when the provided values far exceed 100 g, for all the different SAC-MKPC samples, the added amount of water, potassium dihydrogen phosphate, and gypsum according to the following rules:

  • the M/P was set to 5/1;
  • the water-to-solids (w/s) weight ratio was 16% for the binder, which contained all the MgO and phosphate, and 28% for the SAC clinker;
  • gypsum of 5% mass SAC clinker was added;

  1. As I mentioned above, there are several paragraphs or even chapters which need re-writing, in my opinion.Chapters 3.2 and 3.3 are typical examples, which contain many incomprehensible statements, vague expressions and inaccurate descriptions thus making them very confusing. Minerals are not “visible” (a commonly appearing phrase) in the X-ray diffractograms but only peaks appear, which are interpreted as minerals. I suggest the authors to better describe the XRD and the TG/DTG graphs here and throughout the text. I was lost in several places as there are many repetitions and the appearance and decomposition of several phases is repeated again and again without any clear logic in the text. The authors describe several events on the TG/DTG graphs (actually I suggest to describe them as endothermic or exothermic if any) but most of them are invisible in the panels of Figure 9 (e.g., lines 313-326; all these events do not appear clearly in the graphs and I suggest to indicate them). There are several weird expressions indicating that this (and few other) chapter(s) are written a bit superficially: for example, in line 364 it is written: “…endothermic valley that appears between 60–200℃ was mainly MgKPO4·6H2O…” but it is strange to read that the valley is the phase. The valley indicates the decomposition apparently, which suggests the presence of that phase. There are several places in the text with such a superficial writing.

  1. Response:Thank you so much for your efforts and precious comments on our manuscript.We are sorry for the ambiguous expression in Chapters 3.2 and 3.3. Considering the reviewer’s suggestion, the Chapters 3.2 and 3.3 has been re-writingin the revised manuscript.

We tried our best to improve the Chapters 3.2 and 3.3 and made some changes in the manuscript. These changes will not influence the content and framework of the paper. And here we did not list the changes but marked in red in revised paper. We appreciate for your warm work earnestly, and hope that the correction will meet with approval.

  1. In the same chapter, I cannot follow most of the text included in lines 381-402, as it is carelessly written and contains many inaccuracies. The 30-200 ℃and the 200-700 ℃samples show variations in their mass losses (Table 5). It is not clear that the clinker of 1300 ℃ shows always the highest loss. The losses in the aliquots of the theoretical MgO of 80% at 1300 ℃ and 1350 ℃ are identical whereas for the rest aliquots and the various theoretical MgO values the losses are variable. Likewise, for the 200-700℃ samples. Moreover, I see in Table 5, a value of -0.25 (negative) for the 1250-70% sample. What does it mean? Mass gain? How is it explained? Suddenly, the text in lines 396-402 looks odd here. It appears just out of the blue and out of the context. Generally, this chapter is very confusing and in my view needs rewriting.

1.Response:We are sorry for the ambiguous expression in the chapter. And we are sorry for the careless mistakes in Table 5. As suggested by the reviewer, and the chapter has been re-writing as the following:

Table 5 shows that all samples had an increased mass loss with increasing theoretical MgO values at 30-200°C, but all samples had an decreased mass loss with increasing theoretical MgO values at 200-700°C. At 30–200°C, the MgO theoretical content was 60%–80%, the largest mass loss of the SAC-MKPC hydration products occurred when the clinker was prepared at 1300°C, followed by 1350°C and 1200°C, and the smallest mass loss occurred at 1250°C. Therefore, the sample which the MgO theoretical content was 60%–80% at 1300 °C showed the highest AFt and MgKPO4·6H2O content compared with that in the other three samples. However, the mass loss of SAC-MKPC hydration products when the clinker was prepared at 1200℃ was higher than that prepared at 1250℃.Combined with an analysis of the mineral composition of the SAC-MKPC clinker in Section 3.1, the anhydrate content decreased with an increasing temperature, and its content was highest, which was observed in the sample at 1200 °C,and AFt phase wasn’t generated by 3CaO·3Al2O3·CaSO4 hydration, hence, the hydration product phase at 1250℃ was lower. At 200–700 °C, the largest mass loss of SAC-MKPC hydration products occurred at 1200°C, followed by 1250°C and 1300°C, and the smallest mass loss occurred at 1350°C. The main mass loss was caused by unreacted KH2PO4 decomposition at the same temperature rang.

And we are sorry for the careless mistakes in Table 5,The value of -0.25 for the 1250-70% sample has been revised 0.45. 

And here marked the changes in red in revised paper. We appreciate for your warm work earnestly, and hope that the correction will meet with approval.

  1. Chapter 3.3 is another example of poorly written text. Apart from the confusing and untidy statements several descriptions and interpretations are inaccurate. There is a tremendous error in the EDX spectra (Fig. 10) as the first peak is the Kα line of C and neither K nor Ca is there! Also spectrum A indicates peaks for Mg, Al, S, K, and Ca and I cannot see any Fe, as it is mentioned in line 409. Also descriptions for area B are inaccurate and please double-check the rest spectra. The provided molar ratios are speculative and in my view erroneous. The images are taken in secondary (and not backscattered) electron mode and the samples are not polished, therefore the acquired numerical data are very far from being real. These spectra can only be used as qualitative data and nothing more. It seems like the authors use the acquired analyses and they manipulate the data to fit them in their views for the various phases. Also it is mentioned that the samples are gold-coated but where is the gold peak? Has it been cancelled?

4.Response:We are sorry for the ambiguous expression in Chapters 3.3. Considering the reviewer’s suggestion and the Chapters 3.3 has been re-writing.

However, our working group made a thorough discussion about the EDS spectra (Fig.10), and checked the related literature. We can get the following conclusion: there was significant difference between EDS and EDX, Energy dispersive X-ray spectroscopy (EDS) is a standard method for identifying and quantifying elemental compositions in a very small sample of material (even a few cubic micrometers). In a properly equipped SEM, the atoms on the surface are excited by the electron beam, emitting specific wavelengths of X-rays that are characteristic of the atomic structure of the elements. An energy dispersive detector (a solid-state device that discriminates among X-ray energies) can analyze these X-ray emissions. Appropriate elements are assigned, yielding the composition of the atoms on the specimen surface. This procedure is called energy dispersive X-ray spectroscopy (EDS) and is useful for analyzing the composition of the surface of a specimen. Energy dispersive X-ray spectroscopy (EDX) is an analytical method for analytical or chemical characterization of materials. EDX systems are generally attached to an electron microscopy instrument such as transmission electron microscopy (TEM) or scanning electron microscopy (SEM). EDX analysis gives a spectrum that displays the peaks correlated to the elemental composition of the investigated sample. The first peak is the line of Ca in the EDS spectra, there was no error in the EDS spectra (Fig.10) in manuscript. The related reference as following:

1.Yu.J.C.;Qian.J.S.;Wang.F.;Li.Z.;Jia.X.W.;Preparation and properties of a magnesium phosphate cement with dolomite,Cement and Concrete Research,2020,138, 106235.

Fig. 13. SEM images of the MPC pastes containing ye'elimite at 28 d prepared with the calcined DBG powders: (a) T1200, (b) T1250

We are very sorry for the mistake in the manuscript. Fe was cancelled in the spectrum A with careless, and we has revised.

At the same time, the mineral phase can also be identified by the provided molar ratio of elements, the same research methods and conclusions was present in the reference.  

1.Zhou.B.;Zhu.H.Y.; Xu.S.Y.;Du.G.H.;Shi.S.;Liu.M.; Effect of phosphogypsum on the properties of magnesium phosphate cement paste with low magnesium-to-phosphate ratio, Science of The Total Environment Volume,2021, 798, 1, 149262.

3.6 Microstructure and morphology

The microstructure and morphology of MPPC specimen with 20% and 40% PG was observed by SEM-EDS and presented in Fig. 9. Moreover, the details of EDS analysis are summarised in Table 3. As shown in Fig. 9(a), a small number of prismatic substances with a smooth surface bonded with particles was observed, which is similar to other literature. As shown in Table 3, the component of Area A was mainly O, Mg, P, and K elements with a molar ratio of Mg:P:K around 1:1:1, while for point A, mainly composed of O and Mg elements, the O:Mg:P:K molar ratio is about 1:0.45:0.02:0.01. 

2. Zhang.G.; Li.G.X.; He.T.S.; Effects of sulphoaluminate cement on the strength and water stability of magnesium potassium phosphate cement Construction and Building Materials.2017.132, 1,335-342.

According to Table 4, areas 2 and 3 were present as the main element of the large irregular piece, containing a variety of elements. The molar ratio of elements in areas 2 and 3 are quite similar, for which O:Mg:Ca:C:S was 3.6:0.1:1.0:0.4:0.9 and 4.4:0.1:1.0:0.5:1.1, respectively. New crystals were not detected in the XRD, and this irregular substance must be in the form of gel. Containing O, Ca, and S elements, this substance possibly was the hydration production of anhydrous calcium sulphoaluminate.

Spectrum processing : Peak possibly omitted : 2.849 keV

Processing option : All elements analyzed (Normalised), Number of iterations = 3

Element

Weight%

Atomic%

O K

42.42

61.75

Mg K

3.84

3.68

Al K

0.07

0.06

Si K

0.09

0.08

P K

0.58

0.44

S K

20.24

14.70

K K

18.44

10.98

Ca K

14.28

8.30

Fe K

0.04

0.02

Totals

100.00

It is mentioned that the samples are gold-coated in 2.3. Experimental instruments and testing methods. However, the Au element was not exist in the clinker system, hence, the gold peak has been cancelled in the EDS spectra.

  1. The title of chapter 3.6 does not reflect the contents of the text below. I cannot see any mechanism but the authors merely repeat some chemical reactions from previous sections of the ms and actually a series of reactions that can be found in several other papers, including Ren et al. (2019). What is innovative in this chapter? What does it describe other than a mere reference of some reactions? By the way, this manuscript shows a significant overlap with Ren et al. (2019). How does this ms differ from that paper in terms of the recommended reactions?

1.Response:Thank the reviewer for pointing out this unsuitable title, and we has revised “3.6. Mechanism analysis ” into  “Reaction pathways of mineral formation”. In fact, the paper “Preparation of sulphoaluminate-magnesium potassium phosphate cementitious composite material under low-temperature, Construction and Building Materials,Ren et al.(2019)” was our group’s published paper.This research was mainly focused on investigating the feasibility of using magnesium sulphate (99% MgSO4; Aladdin), calcium oxide (99% CaO; Aladdin), and aluminium oxide(99% Al2O3; Aladdin) in the preparation of the SAC-MKPC composite clinker at 1200-1350℃. However, the submitted manuscript based on the previous paper, industrial wastes (aluminum slag, carbide slag, coal gangue and magnesium desulfurization slag) have been used as materials to produce SAC-MKPC clinker for this research. The industrial wastes have complicated chemical composition, the chemical reaction process was more complicated, meanwhile, the water resistance of SAC-MKPC cementitious composite material has been investigated.

  1. Conclusion No 3: I could not find any discussion in the text about the density of the structure.

1.Response:Thank the reviewer for pointing out the ambiguous expression about the density of the structure in Conclusion No 3. As suggested by the reviewer, we have carefully revised. 

The revised Conclusion 3 as follows:

The XRD analysis of the hydration products of the SAC-MKPC composite indicated that K-struvite and ettringite coexisted. The SEM micrographs also showed that the mutual adhesion of the FAt and K-struvite crystals led to the formation of a very dense structure, The dense structure provided the SAC-MKPC with an excellent water resistance. This novel preparation method could use industrial solid wastes as raw materials to prepare SAC-MKPC high-value cementitious composite material.

7.Line 50: Here and in every citation onwards, please insert a space between the comma and the date and omit semicolon at the end.

Response:Thank you very much for your advice. As you suggest, the all citation has been revised and hope that the correction will meet with Materials reference format.

8.Line 54: Syntax error; please change.

Response:Thank you very much for your advice. We are very sorry for the mistake in the manuscript. As you suggest, the revised part as follows:

A novel approach to improve the water resistance, reduce the use of natural resources and maintain or improve performance MKPC is required.

9.Line 76: Replace “is” with are.

Response:Thank the reviewer for pointing out the inaccurate expressions, we have corrected it.

10.Line 90: Omit “therefore” and replace “material” with materials.

Response:Thank you very much for your advice and we are sorry for the careless mistakes. Considering the reviewer’s suggestion, we have revised the part as follows: 

Because of the release of gaseous ammonia for MgO-NH4H2PO4 system, KH2PO4 (KDP) was used as one of the main raw materials for the SAC-MKPC cementitious composite material preparation.

11.Lines 97-98: Please consider grammar and replace “patterns” with pattern.

Response:Thank the reviewer for pointing out this fault due to our careless and negligence. We have revised the part as follows:

The XRD pattern of magnesium desulfurization slag is shown in Figure 1(a). The magnesium desulfurization slag was mainly composed of magnesium sulfite hexahydrate (MgSO3·6H2O), magnesium sulfate heptahydrate (MgSO4·7H2O), calcium oxide (CaO) and magnesium hydroxide (Mg (OH)2).

12.Lines 100-102: I assume you mean these phases comprise 95% of the material. If yes, please rephrase accordingly. How did you quantify them?

Response:Thank the reviewer for pointing out the ambiguous expression. Considering the reviewer’s suggestion. The revised part as follows:

MgSO3·6H2O, MgSO4·7H2O and Mg (OH)2 are the most important major phases in magnesium desulfurization slag, and these phases comprise 95% percent of the magnesium desulfurization slag material.

Combined XRD pattern and XRF data of desulfurized magnesium slag, the main phases were magnesium sulfate heptahydrate (MgSO4·7H2O), magnesium sulfite hexahydrate (MgSO3·6H2O), calcium oxide (CaO) and magnesium hydroxide (Mg(OH)2), and the content of MgO, SO3,CaO 31.46, 58.15, and 1.26 respectively, the loss on ignition was 6.23. Hence, these phases occupy at least 95% of the material.

13.Lines 106-107: It would be nice to know which exact polymorph of silica, ferric oxide, aluminium oxide and CaCO3 there are. XRD is a good tool to do that and it can largely benefit the manuscript as different polymorphs have different crystal structures, which may help in several interpretations.

Lines 111-112: Fluoride is an anion and cannot be detected with XRD! Also how do you know that all of them are traces? What is the detection limit of the XRD? Generally, the reference of all these materials here is vague.

Response:Thank the reviewer for pointing out the ambiguous expression about the word “Fluoride”, The ““Fluoride” should be calcium fluoride or sodium fluoride. As suggested by the reviewer,

the revised paragraph as follows:

It contains aluminum and aluminum oxide, and may include many impurities, such as sodium chloride, potassium chloride, calcium fluoride, sodium fluoride, aluminum, aluminum nitride, magnesium chlorides and some heavy metals.

The added reference was as follows:

Arunabh,M.;, Kamalesh.K.S.;Recovery of valuable products from hazardous aluminum dross: A review,Resources, Conservation & Recycling,2018, 130,95–108.

14.Line 116: What is “resent”?

Response:Thank the reviewer for pointing out this fault, “resent” is “present”. As suggested by the reviewer,“Some minor phase CaO, TiO2 and potassium salts are resent” has been revised into “Some minor phase CaO, TiO2 and potassium salts are present”.

15.Lines 124-125: Please replace here and throughout the text the term “crystal water” with

crystalline water. Add anhydrous before MgSO3.

Response:Thank you very much for your advice. As you suggest, we have revised the text “crystalline water” and “anhydrous MgSO3” in the revised manuscript.

16.Line 133: I believe you do not have to give a temperature range here because it is confusing. Either the decomposition started at 400 ℃ or the decomposition occurred at this range.

Response:Thank you very much for your advice. As you suggest, we have revised the section as follow:

When the temperature reached 400°C, Ca(OH)2 began to decompose and converted into H2O and CaO, and the decomposition was completed at 500°C in the revised manuscript.

17.Line 135: I see that decomposition is completed at 500 and not 550℃.

Response:Thank the reviewer for pointing out this fault. We have revised the “550℃” into  “500℃” in the revised manuscript.

18.Lines 136-137: Vague statement. Please modify.

Response:Thank the reviewer’s suggestion. The corresponding revision has been done for 136-137, as the following:

Hence, the main quality loss for carbide slag was the decomposition of Ca (OH)2 and CaCO3. 

19.Figure 3. What is the cooling rate or the overall cooling time in the last stage (green line)?

Response:Thank the reviewer’s suggestion. The corresponding revision has been done in the revised manuscript, as the following:

After calcination, the samples were taken out of the furnace and allowed to cool naturally to room temperature in air.

20.Line 243: Replace “mian” with main.

Response:We are very sorry for the mistake in the manuscript. “mian” has been revised into "main".

21.Lines 264-265: Sharp peaks do not indicate the relative amounts of any phase in XRD graphs. At the end it is mentioned: “…the other minerals differed.”. In what terms?

Response:Thank you very much for your advice and we are sorry for the ambiguous expression.Considering the reviewer’s suggestion, we have revised the manuscript as follows:

The final calcination clinker were mainly composed of ye'elimite (), periclase (MgO), and dicalcium silicate (C2S). A small amount of magnesia-aluminum spinel (MgAl2O4), anhydrite (CaSO4), and akermanite (Ca3MgSi2O8) were observed in all samples, which indicated that all Ca(OH)2,MgSO4,MgSO3,Al2O3,CaCO3 and SiO2 in the raw materials were chemically combined after calcination. Moreover, the peak intensities of anhydrite decreased with an increasing calcination temperature.

22.Lines 267-270: Omit “and” before “less mineral-…”; “visible” is an expression frequently used throughout the text for the difractograms but it creates confusion. I recommend to replace “produced” with detected. Omit “mineral phase” (line 270) after ye’elimite. At the end of this sentence please write gradually formed” (not formed gradually) but how do you know that it was gradually? Maybe you can say that ye’elimite appears in the sample prepared at 1300℃.

Response:Thank the reviewer’s suggestion. The corresponding revision has been done for 267-270, as the following:

periclase (MgO), dicalcium silicate (Ca2SiO4), residual anhydrite (CaSO4),less mineral-phase magnesia-alumina spinel (MgAl2O4) and merwinite (Ca3MgSi2O8) was produced in the SAC-MKPC clinker system, but, ye'elimite was not produced. With an increase in calcination temperature, ye’elimite appears in the sample prepared at 1300℃.

23.Lines 270-272: Vague, strange and incorrect sentence. Please re-write.

Response:Thank you very much for your advice and we are sorry for the ambiguous expression. Considering the reviewer’s suggestion, we have revised the manuscript as follows:

The XRD patterns of SAC-MKPC clinker powders with calcination temperatures between 1300 and 1350°C are shown in Fig.6-7. The final calcination products were mainly composed of ye'elimite (), periclase (MgO),and dicalcium silicate (C2S). A small amount of magnesia aluminum spinel (MgAl2O4)  and akermanite (Ca3MgSi2O8) were also found in the calcination products.

24.Line 273: I suggest to omit “phase”.

Response:Thank the reviewer’s suggestion. Considering the reviewer’s suggestion, the “phase” has been deleted.

25.Lines 288-290: Vague, please re-phrase.

General Response:Thank the reviewer’s suggestion. Considering the reviewer’s suggestion,

we have revised the manuscript as follows:

According to the XRD patterns of SAC-MKPC clinker powders with calcination temperatures between 1200 and 1350°C under different raw materials rations , the desired mineral (, MgO) was produced,meanwhile, aluminum spinel (MgAl2O4) and akermanite (Ca3MgSi2O8) were also found in the SAC-MKPC clinker system.

26.Line 292: I suggest to change to gradually prepared; but what do you mean here? How can it be gradually prepared?

Response:Thank you very much for your advice and we are sorry for the ambiguous expression. Considering the reviewer’s suggestion, we have revised the manuscript as follows:

Compared with the magnesium phosphate cementitious material, SAC- MKPC clinker can be prepared from a relatively low-temperature region. The “gradually” has been deleted.

27.Line 298: Please be specific and accurate. This is an X-ray diffractogram and not the mineral composition. “…at a 28-day curing age.” Maybe you mean after 28 days of curing?

Response:Thank the reviewer for pointing out this fault. Considering the reviewer’s suggestion, we have revised the manuscript as follows:

Figure 8 shows the XRD pattern of the SAC-MKPC hydration product after 28 days of curing.

28.Line 304: I suggest to replace with initially achieved.

Response:Thank you very much for your advice. As you suggest, we have replaced obtained initially with initially achieved in the revised manuscript.

29.Line 306: Legend of Fig 8: after 28 days of curing?

Response:Thank you very much for your advice and we are sorry for the ambiguous expression. Considering the reviewer’s suggestion, “XRD pattern of SAC-MKPC paste at 28 days” has been revised into “ XRD pattern of SAC-MKPC paste after 28days of curing”

30.Lines 307-308: Please rephrase. Vague.

Response: Considering the reviewer’s suggestion, we have revised the manuscript as follows:

The TG-DSC results of the SAC-MKPC pastes prepared with calcination temperatures between 1200 and 1350°C cured 28 days are shown in Fig.9.

31.Lines 312-326: As I suggested above, it would be great to indicate all these events of the TG/DTG graphs.

Response:Thank you very much for your advice. Considering the reviewer’s suggestion, we have revised the manuscript as follows:

The TG-DSC results of the SAC-MKPC pastes prepared with calcination temperatures between 1200 and 1350°C cured 28 days are shown in Fig.9. The hydration products of SAC-MKPC included hydrated calcium aluminate sulfate(AFt-ettringite and AFm-monosulfate), MgKPO4·6H2O, aluminum glue, iron glue and calcium silicate hydrate gel according to previous studies [30-32]. Meanwhile, unreacted ye'elimite, MgO, anhydrite and KH2PO4 were also present. Hence, the thermal mass loss of the hydration products of SAC-MKPC can be divided into two main stages. The first stage temperature range is from 30°C to 200°C, the main hydration MgKPO4·6H2O, AFt, iron glue and CaSO4·2H2O lost water. The second stage temperature range is from 30°C to 200°C, unreacted KH2PO4 and aluminum glue lost water.

32.Table 5. Please write the data in this table in consistent letter font. Samples 1200 are different than the lines with samples 1250, 1300, etc.

Response:Thank the reviewer for pointing out this fault in the table 5 due to our careless and negligence. Table 5 has the consistent letter font in revised manuscript.

33.Lines 381-382: Sorry I cannot follow this. Do you mean that the samples show increased mass loss with increasing theoretical MgO values?

Response:Thank you very much for your advice and we are sorry for the ambiguous expression.We have revised the manuscript as follows:

Table 5 shows that all samples had an increased mass loss with increasing theoretical MgO values at the same temperature range.

34.Line 433: Please replace “results: with spectra.

Response: Considering the reviewer’s suggestion, the “results” has been replaced with “spectra”

35.Lins 445-446: I suggest replacing “…and some impurity 445 phases MgAl2O4 and Ca3MgSi2O8.” with and MgAl2O4 and Ca3MgSi2O8 impurities.

Response:Thank you very much for your advice. As you suggest, we have revised the section as follow:

“the main mineral phases of the SAC-MKPC clinker were MgO, CaSO4, a small amount of Ca2SiO4 and some impurity phases MgAl2O4 and Ca3MgSi2O8” has been revised into “the main mineral phases of the SAC-MKPC clinker were MgO, CaSO4, a small amount of Ca2SiO4 and MgAl2O4 and Ca3MgSi2O8 impurities”

36.Line 472: What is changing rules?

Response:Thank you very much for your advices and we are sorry for the ambiguous expression. Considering the reviewer’s suggestion, the

The compressive strength developments of the SAC-MKPC pastes prepared with different theoretical MgO contents at the same calcination temperature at various curing ages are presented in Fig. 12.

37.Lines 473-474: What are these 2h, 1, 3 d etc.? Please define. Here is a typical example (among several others) of poor and superficial writing. Do they refer to curing time, to hydration time or what?

Response:Thank you very much for your advices and we are sorry for the ambiguous expression. Considering the reviewer’s suggestion, the 2h, 1d, 3d, 28d compressive strengths  was the compressive strength of test block at 2h,1day,3day and 28day curing time

38.Lines 475-476: Here and in few places below: “early and later compressive strengths” refer to what?

Response:Thank the reviewer for pointing out the inaccurate expressions. Considering the reviewer’s suggestion, we have revised the manuscript as follows:

The early (from 2h to 3-d) and later (28-d) compressive strengths of the SAC-MKPC increased significantly with an increase in MgO content.

39.Lines 485-488: Please rephrase. Vague. The phrase “with unchanged…” creates confusion although I understand what you mean. I suggest to rephrase it throughout the text.

Response:Thank the reviewer for pointing out the inaccurate expressions.  Considering the reviewer’s suggestion, we have revised the manuscript as follows:

However, when the calcination temperature was unchanged, the theoretical MgO content was 70% and 80% and the 2-h compressive strength and late strength were higher than other theoretical MgO contents. The 3-d compressive strength was close to the 28-d compressive strength, and the later strength was relatively stable.

40.Lines 504-505: The definitions of f and F here are identical! Also be consistent, If you use f in italics it should always appear in italics (e.g. Table 7).

Response:Thank the reviewer for pointing out the inaccurate expressions, we have corrected it.

where K denotes the compressive strength retention rate of the sample, f denotes the compressive strength of the sample at 28 days (MPa) in water and F denotes the compressive

strength of the sample at 28 days (MPa) after air-curing.

41.Figure 13. There is no need for both graphs. Only one is enough. For example, panel a describes perfectly bot the samples of different MgO values (X-axis) and the samples of different M/P ratios (Y-axis). Therefore, the second panel is redundant and unnecessary.

Response:Thank you very much for your advices. Considering the reviewer’s suggestion, we have deleted the Figure 13(b), and the related statement about Figure 13(b) has been revised.

42.The reference list is not in the acceptable format.

Response: Thank the reviewer for pointing out this fault in the reference lists due to our careless and negligence.

 The all reference have been revised according to the Materials reference format in the revised manuscript.

 We tried our best to improve the manuscript and made some changes in the manuscript. These changes will not influence the content and framework of the paper. And here we did not list the changes but marked in red in revised paper. We appreciate for your warm work earnestly, and hope that the correction will meet with approval.

 Once again, special thanks to you for your good comments and suggestions.

Reviewer 2 Report

The manuscript provides an interesting finding from an experimental study of a novel cementitious binder composition as claimed by the authors. However, the proposed mechanism and discussion of the results should be supported in-depth for example by thermodynamic modeling. This has been an established tool to provide the scientific rigor of predicting the reaction products and pore solution chemistry in hydrated cement systems. 

Author Response

Thank you and the reviewers so much for your efforts and precious comments on our manuscript.

After receiving your letter, our working group made a thorough discussion about all the comments and advice. Then, a careful revision was implemented and the details were listed in the following part.

We hope the revised version could meet the publication requirements in Materials. If you have any other concerns, we would like to give further revisions or responses as soon as we can.

Best regards and many thanks!

Reviewer 3 Report

The subject paper is interesting and its purpose complies with the journal’s aim and scope. It presents a great number of experimental results on the combined effect of calcination temperature and a novel SAC-MKPC clinker. The paper is advancing the current knowledge in the field, it is original, the results are sound but need restructuring. The manuscript is suitable for publication after significant improvements.

Lastly, in terms of language the manuscript is well written but requires grammar and language corrections.

In greater detail, the following must be improved:

Abstract:

Calcination temperature, not calcining temperature

The structure of the paper is confusing.

The authors should not mix the paragraphs – materials, methods, results, discussion. For this I suggest:

For the materials and experimental methods section:

  1. Provide a list of all materials used
  2. Provide a list of all methods used separating between raw materials characterization and the characterization of the final combinations

Then the results and discussing section should follow:

  1. When discussing the results, discuss first the raw materials characterization and then, the characterization of the final combinations

Lines 122-137: Please add a reference

Lines 203-204: how much water was used for the cement pastes (I presume by slurry you mean pastes)?

Although in Lines 203-204 you informed us about the formation of “slurry” in table 4 you have assigned “Cement mortar”. Please clarify and add or delete accordingly. Also add the water to binder ratio.

Line 243: main not mian

Figure 4: nice presentation of XRD. Figure 4a the arrow for the MgO phase is not clear. There is something wrong with its direction. The same for 5a. maybe explain the 2θ angle point that you refer too?

Line 441: Vicat is referred to for the first time in the manuscript – please add it in the experimental methods section with all related information

Line 493: water resistance – please add it in the experimental methods section with all related information

Author Response

 Responses to reviewer #2:

Reviewer #2: The subject paper is interesting and its purpose complies with the journal’s aim and scope. It presents a great number of experimental results on the combined effect of calcination temperature and a novel SAC-MKPC clinker. The paper is advancing the current knowledge in the field, it is original, the results are sound but need restructuring. The manuscript is suitable for publication after significant improvements. Lastly, in terms of language the manuscript is well written but requires grammar and language corrections. In greater detail, the following must be improved.
General Response:Thank the reviewer very much for your efforts on our paper and for your approval to the work! According to the reviewer's good instruction, we have revised the whole manuscript carefully and tried to avoid any grammar or syntax error. In addition, the language was also polished by the Elsevier official service, English Language Editing and we reviewed the whole manuscript carefully to avoid language errors. We hope that the language is now acceptable for the next review process. Thank you sincerely for giving us the opportunity to improve our paper to a new level. If you have any other concern, we would like to give further responses as soon as we can.

Specific comments:

1.Abstract: Calcination temperature, not calcining temperature

 Response:Thank you very much for your advice. We are very sorry for the mistake in the Abstract. As you suggest, the “calcining temperature” have been revised into “calcination temperature”.

2.The structure of the paper is confusing. The authors should not mix the paragraphs – materials, methods, results, discussion. For this I suggest: For the materials and experimental methods section:

  1. Provide a list of all materials used
  2. Provide a list of all methods used separating between raw materials characterization and the characterization of the final combinations

Then the results and discussing section should follow:

  1. When discussing the results, discuss first the raw materials characterization and then, the characterization of the final combinations

 Response:Thank the reviewer for pointing out the ambiguous expression about the structure of the paper. Following your comments, we have revised the manuscript and the revised paragraphs as follows:

  1. Table1 has been revised, all the raw materials (aluminum slag, carbide slag, coal gangue, magnesium desulfurization slag -MDS and Secondary added materials-KH2PO4) has been organized  as follows.

Table 1 Main composition of raw materials (wt%).

MgO

Al2O3

SiO2

SO3

CaO

Fe2O3

TiO2

R2Oa

LOIb

SAMc

Carbide slag

0.34

1.33

1.41

1.24

75.05

0.25

0.03

0.21

20.14

KH2PO4(99%,Aladdin)

Aluminum slag

4.87

70.79

9.65

0.41

1.95

4.11

0.49

3.51

4.22

Coal gangue

2.55

20.16

61.62

1.95

2.27

3.28

1.24

0.71

6.22

MDS

31.46

1.21

1.03

58.15

1.26

0.31

0.12

0.23

6.23

a Alkaline oxide (K2O, Na2O) ; b Loss on ignition at 950℃; c Secondary added materials 

2.Thank you very much for your advice. As you suggest, the main methods of raw materials, clinker and hydration products has been list as follows. 

Table 2 Main methods of raw materials, clinker and hydration products.

TG-DTG

XRF

XRD

SEM-EDS

Setting time

Compressive strength

Water resistance

Raw materials

Clinker

Hydration products

3.Thank you very much for your advice. Our working group made a thorough discussion about all the structure of the paper. We think that the raw materials characterization should not be revised into the part of Results and discussion. The main reasons as follow:

  • The control curve of calcination temperature was based on the mineral composition of raw materials, first of all, the mineral composition and TG-DTG analysis of raw materials was invested, and then the control curve of calcination temperature can be formed.

(2)The SAC-MKPC cementitious composite material required secondary mixing with SAC-MKPC clinker, gypsum and KH2PO4. The MgO content of SAC-MKPC clinker affect the added content of KH2PO4.Hence,

Considering the reviewer’s suggestion, the structure of the paper has been revised, the chapter 2.3 has been revised as the first paragraph in Section 2.

3.Lines 122-137: Please add a reference

Response:Thank you very much for your advice and the related reference was added.

Reference:

Kou,R.; Guo,M.Z.; Han,L.; Li,J.S.; Li,B.; Chu,H.Q.Recycling sediment, calcium carbide slag and ground granulated blast-furnace slag into novel and sustainable cementitious binder for production of eco-friendly mortar. Constr. Build. Mater.2021,305, 25, 124772.

4.Lines 203-204: how much water was used for the cement pastes (I presume by slurry you mean pastes)?

Response:Thank you very much for your advices and we are sorry for the ambiguous expression about the water used and slurry. However, for each SAC-MKPC clinker sample, the MgO content and ye'elimite content are different, hence, the water used is different, for all the different SAC-MKPC samples, the added amount of water, potassium dihydrogen phosphate, and gypsum according to the following rules:

  • the M/P was set to 5/1;
  • the water-to-solids (w/s) weight ratio was 16% for the binder, which contained all the MgO and phosphate, and 28% for the SAC clinker;
  • gypsum of 5% mass SAC clinker was added;

5.Although in Lines 203-204 you informed us about the formation of “slurry” in table 4 you have assigned “Cement mortar”. Please clarify and add or delete accordingly. Also add the water to binder ratio.

Response:Thank you very much for your advices and we are sorry for the careless mistakes. Considering the reviewer’s suggestion, we have revised the “slurry” into “paste”. 

“Cement mortar vibration table” was just an experimental instrument which can vibrate the paste in the mold. The cement mortar and cement paste can be vibrated with this instrument. The water to binder ratio has been answered in the previous question.

Cement mortar vibration table

6.Line 243: main not mian

Response:We are very sorry for the mistake in the manuscript. “mian” has been revised into "main".

7.Figure 4: nice presentation of XRD. Figure 4a the arrow for the MgO phase is not clear. There is something wrong with its direction. The same for 5a. maybe explain the 2θ angle point that you refer too?

Response: Thank you very much for your precious advice. Following your comments, we revised the Figure 4a.We hope the revised Figure 4a could meet your requirements. Figure 4b refers to the minor phase of SAC-MKPC clinker. If you have any other suggestions, we'll try our best to revised them.

8.Line 441: Vicat is referred to for the first time in the manuscript – please add it in the experimental methods section with all related information

Response:Thank you very much for your advices and the Vicat instrument information has been added in Table 4 and the experimental methods section, the revised experimental methods section additions as follows:

At an ambient temperature of 25°C, the setting time of the fresh SAC-MKPC paste was measured with automatic setting time tester according to GB/T1346-2011. Since the time between the initial and final setting was very short (several minutes), only the initial setting time was presented in this paper.

9.Line 493: water resistance – please add it in the experimental methods section with all related information

Response:Thank you very much for your advices and the water resistance experimental method added in the experimental methods section, the revised experimental methods section additions as follows:

The compressive strength of the samples was measured using an automatic pressure measurement testing machine. Each proportion was measured as a set of two samples. The samples after air-curing for 2h, 1 day, 3 day, 7 days, and 28 days were tested for compressive strength. For testing water resistance, the samples after 28 days in water, the compressive strength was measured. Before testing, the samples immersed in water were taken out of the water and dried for 4 h. 

We tried our best to improve the manuscript and made some changes in the manuscript. These changes will not influence the content and framework of the paper. And here we did not list the changes but marked in red in revised paper. We appreciate for your warm work earnestly, and hope that the correction will meet with approval.

 Once again, special thanks to you for your good comments and suggestions.

Round 2

Reviewer 1 Report

Dear Authors

Thanks for the revised version and most of them have been adequately addressed. However, I have a major problem and I firmly insist on the Comment No 4. I'm afraid you have confused some things. EDX and EDS are exactly the same and even in Wikipedia you can see that. The first peak in an EDX or EDS spectrum is undoubtedly carbon and you may visit several reliable sites to confirm (e.g. https://www.globalsino.com/EM/page1853.html; https://www.microscopy.ethz.ch/xray_spectrum.htm). If you have found it as Ca in any other paper it is erroneous and please stop propagating the error. I also insist that you cannot calculate any molar ratios from the EDX peaks, as they are only qualitative and not quantitative. Also please note that the peaks at the left side (smaller keV) are more effective and hence elements at the same composition appear with higher peaks than the higher keV side.

In my opinion, the text require more linguistic polishing but I leave the final decision on this with the Editor

Reviewer 2 Report

The manuscript has improved.

Reviewer 3 Report

The authors have effectively addressed the comments. 

The manuscript is now suitable for publication.